# PROBABILISTIC VARIATIONAL CONTRASTIVE LEARNING

## ABSTRACT

Deterministic embeddings learned by contrastive learning (CL) methods such as SimCLR and SupCon achieve state-of-the-art performance but lack a principled mechanism for uncertainty quantification. We propose *Variational Contrastive Learning* (VCL), a decoder-free framework that maximizes the evidence lower bound (ELBO) by interpreting the InfoNCE loss as a surrogate reconstruction term and adding a KL divergence regularizer to a uniform prior on the unit hypersphere. We model the approximate posterior $q_\theta(z|x)$ as a projected normal distribution, enabling the sampling of probabilistic embeddings. Our two instantiations—VSimCLR and VSupCon—replace deterministic embeddings with samples from $q_\theta(z|x)$ and incorporate a normalized KL term into the loss. Experiments on multiple benchmarks demonstrate that VCL mitigates dimensional collapse, enhances mutual information with class labels, and matches or outperforms deterministic baselines in classification accuracy, all the while providing meaningful uncertainty estimates through the posterior model. VCL thus equips contrastive learning with a probabilistic foundation, serving as a new basis for contrastive approaches.

## 1 INTRODUCTION

Deep representation learning seeks to map each input $x$ into a compact embedding $z$ that preserves semantic similarity and facilitates downstream tasks such as classification or retrieval (Bengio et al., 2013). Contrastive learning methods, including SimCLR (Chen et al., 2020) and SupCon (Khosla et al., 2020), have advanced the state of the art by pulling together positive pairs and pushing apart negatives in the embedding space. However, these approaches rely on deterministic point estimates for each sample, which do not express uncertainty or capture multiple plausible representations.

To address this limitation, we introduce a probabilistic *Variational Contrastive Learning* (VCL) approach, which extends deterministic embeddings to *probabilistic embeddings* by maximizing the evidence lower bound (ELBO) within the contrastive learning framework. Unlike variational autoencoders (VAEs) (Kingma et al., 2019), which employ a decoder to reconstruct inputs from latent variables, VCL omits explicit decoders. Instead, we show that the InfoNCE loss can serve as a surrogate for the ELBO reconstruction term, yielding a principled probabilistic formulation of contrastive learning. Our VCL framework offers several new perspectives on learned embeddings:

**Understanding embeddings through distributions.** VCL maps each input $x$ to an approximate posterior distribution $q_\phi(z|x)$, yielding a mean vector that serves as the embedding and a variance that quantifies uncertainty. This probabilistic representation not only captures richer information about each sample but also enables uncertainty-aware downstream decisions.

**A probabilistic ELBO viewpoint beyond mutual information.** Minimizing the InfoNCE loss maximizes a lower bound on mutual information, consistent with the InfoMax principle (Linsker, 1988). However, mutual information alone may not directly correlate with downstream task performance (Tschannen et al., 2020), motivating geometric analyses based on alignment and uniformity (Wang & Isola, 2020). In contrast, our probabilistic formulation interprets contrastive learning through the ELBO: maximizing the ELBO without a decoder encourages the learned posterior $q_\phi(z|x)$ to match the true posterior $p(z|x)$, providing a principled objective for representation learning.

**Controllable embedding distributions.** Standard contrastive learning imposes no explicit prior on embeddings, so their distribution emerges implicitly from the data and model architecture. VCL incorporates a prior $p(z)$ into the objective, allowing one to specify and control the marginal embedding distribution $p(z)$. For example, choosing a uniform prior on the hypersphere often improves empirical performance (Wang & Isola, 2020) that makes negative samples more uniformly dispersed around each anchor, and domain-specific priors can encode known structure in the data.

**Mitigating collapse phenomena.** Self-supervised contrastive learning (e.g., SimCLR) can suffer from *dimensional collapse*, where embeddings occupy only a low-dimensional subspace (Jing et al., 2021). A spherical uniform prior mitigates this by encouraging isotropic use of all embedding dimensions.

This Variational Contrastive Learning framework thus provides uncertainty-aware embeddings, a new basis of CL with theoretical insights via the ELBO, and practical solutions to common collapse problems in contrastive learning. Our contributions are summarized as follows:

- We introduce *Variational Contrastive Learning* (VCL), a decoder-free ELBO maximization framework that reinterprets the InfoNCE loss as a surrogate reconstruction term and incorporates a KL divergence regularizer to a uniform prior on the unit hypersphere.

- We propose a distributional embedding model using a projected normal posterior $q_\theta(z|x)$ that enables sampling, uncertainty quantification, and efficient KL computation on the hypersphere.

- We derive a theoretical connection between the optimal InfoNCE critic and the ELBO, showing that minimizing InfoNCE asymptotically maximizes the ELBO reconstruction term (Proposition 3.2) and providing a generalization bound (Theorem 3.3) of KL regularization.

- We demonstrate that VCL mitigates both dimensional collapse in self-supervised contrastive learning via the KL regularizer, while preserving embedding structure. We show that VCL methods preserve or improve mutual information with labels, match or exceed classification accuracy of deterministic baselines, and provide meaningful implication of distributional embeddings.

## 2 PRELIMINARIES

Let $\mathcal{D} = \{(x_i, y_i)\}_{i=1}^N$ be a dataset of input $x \in \mathcal{X}$ and label pairs drawn i.i.d. from the joint distribution $p(x, y)$. An *encoder* $f_\theta \colon \mathcal{X} \to \mathbb{R}^d$, parameterized by $\theta$, maps each input $x$ to a $d$-dimensional vector, which we then normalize to unit length: $z = \frac{f_\theta(x)}{\|f_\theta(x)\|_2}$. Throughout this section, we define the temperature–scaled cosine similarity between embeddings $z_i$ and $z_j$ as

$$s(z_i, z_j) = \frac{z_i^\top z_j}{\tau}, \tag{1}$$

where $\tau > 0$ is the temperature hyperparameter. For any two probability distributions $q$ and $p$, we denote the Kullback–Leibler (KL) divergence by $D(q \,\|\, p) = \mathbb{E}_{z \sim q}\left[\log \frac{q(z)}{p(z)}\right]$.

### 2.1 SELF-SUPERVISED CONTRASTIVE LEARNING

Self-supervised contrastive learning (SSCL) learns representations from *unlabeled* data by pulling together embeddings of semantically related views (positives) and pushing apart those of unrelated views (negatives). For an anchor $x$, let $x_i'$ denote a positive view sampled from $p(x_i' \mid x)$, and let $\{x_j'\}_{j \neq i}$ be $N - 1$ negative views drawn i.i.d. from the marginal $p(x')$. The InfoNCE loss (Oord et al., 2018) for anchor $x$ is then

$$I_{\mathrm{NCE}}(x; x') = -\mathbb{E}_{\substack{x \sim p(x) \\ x_i' \sim p(x_i'|x) \\ \{x_j'\}_{j \neq i} \sim p(x')}} \left[ \log \frac{\exp\big(s(z, z_i')\big)}{\sum_{j=1}^N \exp\big(s(z, z_j')\big)} \right], \tag{2}$$

where $z = f_\theta(x)/\|f_\theta(x)\|_2$ and $s(\cdot, \cdot)$ is the temperature-scaled cosine similarity.

In practice, following SimCLR (Chen et al., 2020), we generate positives by applying two random augmentations $t', t'' \sim \mathcal{T}$ to each sample $\boldsymbol{x}_i$, yielding $(\boldsymbol{x}_i', \boldsymbol{x}_i'') = (t'(\boldsymbol{x}_i), t''(\boldsymbol{x}_i))$.[1] All other $2N - 2$ augmented samples in the mini-batch serve as negatives. Let $\mathcal{B}$ be the set of all $2N$ embeddings in the batch; then InfoNCE can be computed as

$$I_{\text{NCE}} = -\frac{1}{2N} \sum_{\boldsymbol{z} \in \mathcal{B}} \log \frac{\exp\big(s(\boldsymbol{z}, \boldsymbol{z}_p)\big)}{\sum_{\boldsymbol{z}_n \in \mathcal{B} \setminus \{\boldsymbol{z}\}} \exp\big(s(\boldsymbol{z}, \boldsymbol{z}_n)\big)}, \tag{3}$$

where $\boldsymbol{z}_p$ denotes the positive embedding corresponding to $\boldsymbol{z}$. Since InfoNCE lower-bounds the mutual information $I(\boldsymbol{x}; \boldsymbol{x}')$ via $I(\boldsymbol{x}; \boldsymbol{x}') \geq \log N - I_{\text{NCE}}(\boldsymbol{x}; \boldsymbol{x}')$, we can see that minimizing $I_{\text{NCE}}$ encourages encoders to preserve the semantic information of $\boldsymbol{x}$ (Poole et al., 2019).

## 2.2 Supervised Contrastive Learning

Khosla et al. (2020) extend the InfoNCE loss from the self-supervised setting to a supervised context, calling the resulting method *Supervised Contrastive Learning* (SupCon). When class labels $y_i \in \{1, \ldots, C\}$ are available, all samples sharing the same label can serve as positives.

Given a mini-batch $\{(\boldsymbol{x}_i, y_i)\}_{i=1}^B$, define for each anchor index $a$

$$\mathcal{A}(a) = \{1, 2, \ldots, B\} \setminus \{a\}, \text{ and } \mathcal{P}(a) = \{p \in \mathcal{A}(a) : y_p = y_a\},$$

so that $\mathcal{P}(a)$ contains the indices of all positives for anchor $a$. The SupCon loss for anchor $\boldsymbol{x}_a$ is then

$$I_{\text{SUP}}(\boldsymbol{x}_a) = -\frac{1}{|\mathcal{P}(a)|} \sum_{p \in \mathcal{P}(a)} \log \frac{\exp\big(s(\boldsymbol{z}_a, \boldsymbol{z}_p)\big)}{\sum_{j \in \mathcal{A}(a)} \exp\big(s(\boldsymbol{z}_a, \boldsymbol{z}_j)\big)}. \tag{4}$$

Averaging over all anchors in the batch yields the full objective:

$$\mathcal{L}^{\text{sup}} = \frac{1}{B} \sum_{a=1}^B I_{\text{SUP}}(\boldsymbol{x}_a). \tag{5}$$

## 2.3 Variational Inference and the Evidence Lower Bound (ELBO)

In variational inference (Blei et al., 2017; Kingma et al., 2019), we treat the data distribution $p(\boldsymbol{x})$ as the marginal of a joint distribution over observed data $\boldsymbol{x}$ and latent variables $\boldsymbol{z}$, i.e., $p(\boldsymbol{x}) = \int p(\boldsymbol{x}|\boldsymbol{z}) \, p(\boldsymbol{z}) \mathrm{d}\boldsymbol{z}$. The latent variable $\boldsymbol{z}$ captures meaningful structure in $\boldsymbol{x}$, serving both as a hidden cause and as a compressed representation for downstream tasks. In representation learning, we interpret $\boldsymbol{z}$ as the embedding of $\boldsymbol{x}$.

The log-evidence can be written with respect to any approximate posterior $q_\phi(\boldsymbol{z}|\boldsymbol{x})$ as

$$\log p(\boldsymbol{x}) = \log \mathbb{E}_{q_\phi(\boldsymbol{z}|\boldsymbol{x})}\Big[\tfrac{p(\boldsymbol{x},\boldsymbol{z})}{q_\phi(\boldsymbol{z}|\boldsymbol{x})}\Big]. \tag{6}$$

Rather than optimizing (6) directly, variational methods maximize the *evidence lower bound* (ELBO) obtained as a result of applying Jensen's inequality:

$$\log p(\boldsymbol{x}) \geq \mathbb{E}_{q_\phi(\boldsymbol{z}|\boldsymbol{x})}\big[\log p(\boldsymbol{x}|\boldsymbol{z})\big] - D\big(q_\phi(\boldsymbol{z}|\boldsymbol{x}) \,\|\, p(\boldsymbol{z})\big) \; = \; \mathcal{L}^{\text{ELBO}}(\phi), \tag{7}$$

where $p(\boldsymbol{z})$ is a fixed prior (commonly $\mathcal{N}(\mathbf{0}, I_d)$). The ELBO decomposes into a *reconstruction* term $\mathbb{E}_q[\log p(\boldsymbol{x}|\boldsymbol{z})]$ and a *regularizer* $D\big(q_\phi(\boldsymbol{z}|\boldsymbol{x}) \,\|\, p(\boldsymbol{z})\big)$. Maximizing $\mathcal{L}^{\text{ELBO}}$ thus balances (i) accurate reconstruction, (ii) posterior-to-prior regularization, and (iii) posterior accuracy. By

$$\log p(\boldsymbol{x}) = \mathcal{L}^{\text{ELBO}}(\phi) + D\big(q_\phi(\boldsymbol{z}|\boldsymbol{x}) \,\|\, p(\boldsymbol{z}|\boldsymbol{x})\big), \tag{8}$$

for fixed $\log p(\boldsymbol{x})$, maximizing the ELBO minimizes the KL divergence between the approximate and true posteriors (Blei et al., 2017).

The ELBO provides a tractable surrogate for marginal likelihood that can be optimized by standard gradient methods. It will serve as the theoretical backbone of our Variational Contrastive Learning framework, offering both a probabilistic interpretation and explicit control over latent uncertainty.

---

[1] Although we adopt the SimCLR augmentation scheme, our method applies to any contrastive framework.

**Relation to contrastive objectives.** Although the ELBO stems from latent-variable modeling, its two components align naturally with contrastive objectives: the KL divergence term enforces *uniformity* in the embedding space, while the reconstruction term promotes *alignment* between embeddings and observations. In Section 3, we leverage this connection by adopting distributional embeddings in the contrastive framework and incorporating a KL-based regularizer on the posterior.

# 3 VARIATIONAL CONTRASTIVE LEARNING (VCL)

Unlike existing variational contrastive learning methods—which primarily focus on generative models with explicit decoders (Chen et al., 2025; Wang et al., 2024b)—our approach performs *decoder-free* ELBO maximization, making VCL a truly contrastive learning framework.

## 3.1 DECODER-FREE ELBO MAXIMIZATION

Here we describe how to optimize two terms in ELBO (7) within a purely contrastive learning setup.

**Reconstruction term.** The reconstruction term $\mathbb{E}_{q_\theta(\boldsymbol{z}|\boldsymbol{x})}\big[\log p(\boldsymbol{x}|\boldsymbol{z})\big]$ requires the true conditional $p(\boldsymbol{x}|\boldsymbol{z})$, which is generally intractable. Instead, we approximate it via the embedding conditional

$$p(\boldsymbol{z}'|\boldsymbol{z}) = \frac{p(\boldsymbol{z}, \boldsymbol{z}')}{\int p(\boldsymbol{z}, \boldsymbol{z}')\, \mathrm{d}\boldsymbol{z}'}, \tag{9}$$

where $\boldsymbol{z}' \sim q_\theta(\cdot \mid \boldsymbol{x})$ captures semantics of $\boldsymbol{x}$. Thus,

$$\mathbb{E}_{q_\theta(\boldsymbol{z}|\boldsymbol{x})}\big[\log p(\boldsymbol{x}|\boldsymbol{z})\big] \approx \mathbb{E}_{q_\theta(\boldsymbol{z}|\boldsymbol{x})q_\theta(\boldsymbol{z}'|\boldsymbol{x})}\big[\log p(\boldsymbol{z}'|\boldsymbol{z})\big]$$

$$= \mathbb{E}\Big[\log \frac{p(\boldsymbol{z}, \boldsymbol{z}')}{\int p(\boldsymbol{z}, \boldsymbol{z}')\, \mathrm{d}\boldsymbol{z}'}\Big] \approx \mathbb{E}\Big[\log \frac{e^{\psi(\boldsymbol{z}, \boldsymbol{z}')}}{\sum_j e^{\psi(\boldsymbol{z}, \boldsymbol{z}'_j)}}\Big], \tag{10}$$

where we approximate $p(\boldsymbol{z}, \boldsymbol{z}') \approx e^{\psi(\boldsymbol{z}, \boldsymbol{z}')}$ via a critic $\psi$. Details on parameterizing $p(\boldsymbol{z}' \mid \boldsymbol{z})$ appear in Section 3.2. The following lemma supports the approximation $\mathbb{E}_{q_\theta(\boldsymbol{z}|\boldsymbol{x})}\big[\log p(\boldsymbol{x}|\boldsymbol{z})\big] \approx \mathbb{E}_{q_\theta(\boldsymbol{z}|\boldsymbol{x})q_\theta(\boldsymbol{z}'|\boldsymbol{x})}\big[\log p(\boldsymbol{z}'|\boldsymbol{z})\big]$. A further discussion on the approximation in (10) and a tightness condition is in Appendix C.1.

**Lemma 3.1.** *Let $\boldsymbol{x}$ and $\boldsymbol{z}$ be conditionally independent given $\boldsymbol{z}'$. Then, the reconstruction term in Section 3.1 is bounded as*

$$\mathbb{E}_{q(\boldsymbol{z}|\boldsymbol{x})}[\log p(\boldsymbol{x}|\boldsymbol{z})] \geq \mathbb{E}_{q(\boldsymbol{z}|\boldsymbol{x})q(\boldsymbol{z}'|\boldsymbol{x})}[\log p(\boldsymbol{z}'|\boldsymbol{z})] + \mathrm{const.}, \tag{11}$$

*where* const. *is independent of $\boldsymbol{z}$.*

*Proof.* The proof of Proposition 3.1 is in Appendix B.1. □

Noting that the right-hand side of (10) is (up to sign) the InfoNCE surrogate, setting $\psi(\cdot, \cdot) = s(\cdot, \cdot)$ in (10) where $s(\cdot, \cdot)$ is defined in (1) yields

$$\mathbb{E}_{q_\theta(\boldsymbol{z}|\boldsymbol{x})}\big[\log p(\boldsymbol{x}|\boldsymbol{z})\big] \approx -I_{\mathrm{NCE}}(\boldsymbol{x}; \boldsymbol{x}'). \tag{12}$$

Hence, minimizing the InfoNCE loss maximizes the reconstruction term without explicit decoders.

In contrast to VAE embeddings—which often rely on pixel-level reconstruction through expressive decoder (Song et al., 2024)—VCL preserves semantics via contrastive objectives. The next proposition (proved in Appendix B.2) provides a theoretical connection between InfoNCE and the reconstruction term.

**Proposition 3.2.** *Assume that: 1) the critic $\psi$ in InfoNCE is optimal; 2) $p(\boldsymbol{z}) < \infty$, $\forall \boldsymbol{z}$; and 3) $0 < \epsilon \leq p(\boldsymbol{z}|\boldsymbol{z}') \leq g_+(\boldsymbol{z})$, $\forall \boldsymbol{z}, \boldsymbol{z}'$ with a absolutely integrable $g : \mathcal{Z} \to (0, \infty)$. Then, as the number of negatives $N \to \infty$,*

$$-I_{\mathrm{NCE}}(\boldsymbol{x}; \boldsymbol{x}') + \log N \longrightarrow \mathbb{E}\big[\log p(\boldsymbol{z}'|\boldsymbol{z})\big] - D\big(q_\theta(\boldsymbol{z}'|\boldsymbol{x}) \,\|\, p(\boldsymbol{z}')\big) - H\big(q_\theta(\boldsymbol{z}'|\boldsymbol{x})\big), \tag{13}$$

*where the expectation is over $q_\theta(\boldsymbol{z}|\boldsymbol{x})q_\theta(\boldsymbol{z}'|\boldsymbol{x})$, and $H(\cdot)$ denotes entropy.*

**Regularization.** Maximizing the ELBO requires choosing a prior $p(z)$ and an approximate posterior $q_\theta(z \mid x)$. Although both are often taken as Gaussian distributions (Kingma et al., 2019), this choice conflicts with the geometry of contrastive embeddings, which often lie on the unit hypersphere due to the normalization (Wang & Isola, 2020). Instead, we adopt non-Gaussian priors and posteriors—one key distinction from standard VAE approaches.

Motivated by the uniformity property (Wang & Isola, 2020) on the unit sphere $\mathcal{S}^{d-1} = \{z \in \mathbb{R}^d : \|z\|_2 = 1\}$, we set the prior $p(z)$ to be the uniform distribution over $\mathcal{S}^{d-1}$. For the approximate posterior, we use the *projected normal* distribution (Hernandez-Stumpfhauser et al., 2017), which admits efficient KL-divergence computation while enforcing $z \in \mathcal{S}^{d-1}$. A random variable $z \sim \mathcal{PN}(\mu, K)$ is obtained by sampling

$$z = \frac{\mathbf{u}}{\|\mathbf{u}\|_2} \quad \text{with} \quad \mathbf{u} \sim \mathcal{N}(\mu, K). \tag{14}$$

In particular, $\mathcal{PN}(0, I_d)$ reduces to the uniform distribution on $\mathcal{S}^{d-1}$, i.e., $\mathcal{PN}(0, I_d) \stackrel{d}{=} \mathrm{Unif}(\mathcal{S}^{d-1})$.

With $q_\theta(z|x) = \mathcal{PN}(\mu, K)$, the regularization term becomes

$$D\big(q_\theta(z|x) \,\|\, p(z)\big) = D\big(\mathcal{PN}(\mu, K) \,\|\, \mathrm{Unif}(\mathcal{S}^{d-1})\big). \tag{15}$$

Since a closed-form KL divergence between projected normals and the uniform sphere is intractable, we instead minimize the Gaussian KL as an upper bound—by the data processing inequality (Polyanskiy & Wu, 2025):

$$D\big(\mathcal{N}(\mu, K) \,\|\, \mathcal{N}(0, I_d)\big) \ \geq \ D\big(\mathcal{PN}(\mu, K) \,\|\, \mathrm{Unif}(\mathcal{S}^{d-1})\big). \tag{16}$$

In Appendix C.2, we analyze the tightness of the gap in (16) and show that the Gaussian KL divergence closely approximates the projected-normal KL divergence; the two exhibit nearly identical behavior throughout VCL training.

For $K = \mathrm{diag}(\sigma_1^2, \ldots, \sigma_d^2)$, the Gaussian KL admits the closed form

$$D(\mu, K) = \frac{1}{2} \sum_{i=1}^{d} \big(\sigma_i^2 + \mu_i^2 - 1 - \log \sigma_i^2\big). \tag{17}$$

The KL divergence term $D(\mu, K)$ grows linearly with the embedding dimension $d$, which can destabilize training when $d$ is large. To address this, we normalize the KL term by $d$, i.e., $\widetilde{D}(\mu, K) = \frac{1}{d} D(\mu, K)$, so that its magnitude remains comparable to the InfoNCE loss.

**Final objective for maximizing ELBO.** By combining (12) and (17), we obtain the following (approximate) lower bound on the ELBO:

$$\mathcal{L}^{\mathrm{ELBO}}(\theta) \ \geq \ -I_{\mathrm{NCE}}(x; x') \ - \ D\big(\mu_x, K_x\big), \tag{18}$$

where $\mu_x$ and $K_x = \mathrm{diag}(\sigma_{x,1}, \ldots, \sigma_{x,d})$ are the parameters of $q_\theta(z \mid x)$. Because this bound is asymmetric in $(x, x')$, we symmetrize it to define our final VCL objective:

$$\mathcal{L}^{\mathrm{VCL}} = \frac{1}{2} \Big( I_{\mathrm{NCE}}(x; x') + I_{\mathrm{NCE}}(x'; x) + D(\mu_x, K_x) + D(\mu_{x'}, K_{x'}) \Big). \tag{19}$$

Minimizing $\mathcal{L}^{\mathrm{VCL}}$ therefore maximizes the ELBO. Next, we introduce Variational SimCLR (VSim-CLR), which is specifically designed to optimize this objective efficiently.

### 3.2 VARIATIONAL SIMCLR (VSIMCLR)

We propose *Variational SimCLR* (VSimCLR), whose architecture is illustrated in Figure 1(b). VSim-CLR minimizes $\mathcal{L}^{\mathrm{VCL}}$ in (19), thereby implicitly maximizing the ELBO and bringing the approximate posterior closer to the true posterior by (8). Compared to SimCLR, VSimCLR differs in three key aspects: (i) the encoder outputs the parameters of a variational posterior rather than deterministic embeddings; (ii) embeddings are sampled from this posterior; and (iii) a KL divergence term between the approximate posterior and the prior is included in the loss.

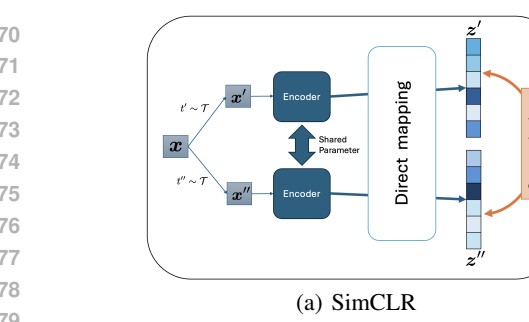
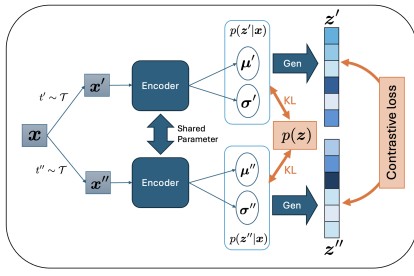

(a) SimCLR                    (b) Variational SimCLR

Figure 1: Graphical illustration of SimCLR and Variational SimCLR (VSimCLR).

Specifically, during training, each input $\boldsymbol{x}$ is first augmented twice to obtain $\boldsymbol{x}'$ and $\boldsymbol{x}''$, as in SimCLR. The encoder then maps $\boldsymbol{x}'$ and $\boldsymbol{x}''$ to posterior parameters $(\boldsymbol{\mu}', \boldsymbol{\sigma}')$ and $(\boldsymbol{\mu}'', \boldsymbol{\sigma}'')$, respectively. We then sample

$$\boldsymbol{z}' = \boldsymbol{\mu}' + \mathrm{diag}(\boldsymbol{\sigma}')\,\boldsymbol{\epsilon}_1, \quad \text{and} \quad \boldsymbol{z}'' = \boldsymbol{\mu}'' + \mathrm{diag}(\boldsymbol{\sigma}'')\,\boldsymbol{\epsilon}_2, \tag{20}$$

where $\boldsymbol{\epsilon}_1, \boldsymbol{\epsilon}_2 \overset{\text{i.i.d}}{\sim} \mathcal{N}(\mathbf{0}, I_d)$. After normalizing $\mathbf{z}'$ and $\mathbf{z}''$ to unit length, we compute the InfoNCE loss over the normalized embeddings in the batch and add the KL divergence

$$\frac{1}{d} D\big(\mathcal{N}(\boldsymbol{\mu}, \mathrm{diag}(\boldsymbol{\sigma}^2)) \,\|\, \mathcal{N}(\mathbf{0}, I_d)\big) \tag{21}$$

for each sample. Minimizing this combined objective effectively minimizes $\mathcal{L}^{\mathrm{VCL}}$ in (19) and thus maximizes the ELBO. Figure 1 highlights these differences: VSimCLR replaces deterministic embeddings with the projected-normal posterior $\mathcal{PN}(\boldsymbol{\mu}, \mathrm{diag}(\boldsymbol{\sigma}^2))$ and regularizes it via KL divergence to the standard normal.

### 3.3 VARIATIONAL SUPCON (VSUPCON)

Building on the variational embedding pipeline of VSimCLR, VSupCon simply swaps the unsupervised InfoNCE term for the supervised contrastive loss while retaining the KL regularizer. Concretely, for each input $\boldsymbol{x}$ with two augmentations $\boldsymbol{x}', \boldsymbol{x}''$, let the encoder output posterior parameters $(\boldsymbol{\mu}', K')$ and $(\boldsymbol{\mu}'', K'')$, and sample normalized embeddings

$$\boldsymbol{z}' \sim \mathcal{PN}(\boldsymbol{\mu}', K'), \qquad \boldsymbol{z}'' \sim \mathcal{PN}(\boldsymbol{\mu}'', K''). \tag{22}$$

Then the VSupCon objective is the symmetrized supervised loss plus the averaged, normalized KL penalties:

$$\mathcal{L}^{\mathrm{VSup}} = \frac{1}{2}\Big(\mathcal{L}^{\mathrm{sup}}(\boldsymbol{z}', \boldsymbol{z}'') + \mathcal{L}^{\mathrm{sup}}(\boldsymbol{z}'', \boldsymbol{z}')\Big) \;+\; \frac{1}{2d}\Big(D(\boldsymbol{\mu}', K') + D(\boldsymbol{\mu}'', K'')\Big). \tag{23}$$

Minimizing $\mathcal{L}^{\mathrm{VSup}}$ therefore aligns same-class embeddings and regularizes their posterior distributions toward the uniform prior on the sphere.

### 3.4 GENERALIZATION ANALYSIS FOR KL DIVERGENCE

Generalization bounds quantify how a loss function performs on unseen data. While recent work has extensively studied the InfoNCE loss (Saunshi et al., 2019; Lei et al., 2023; Hieu et al., 2025), the KL regularizer in VCL has not yet received comparable theoretical treatment. To address this gap, we derive a generalization bound for the KL term under the deep neural network encoder function class $\mathcal{F}_{\boldsymbol{\Xi}}$, parameterized by a bounded weight set $\boldsymbol{\Xi}$ and equipped with Lipschitz activation functions, as considered in the recent work (Hieu et al., 2025). We introduce an informal version of the generalization bound below; the formal version is provided in Theorem B.1 in Appendix B.3.

**Theorem 3.3** (Informal). *Let $\{\boldsymbol{x}_i\}_{i=1}^N \overset{\text{i.i.d}}{\sim} p(\boldsymbol{x})$, and let $D_{\mathrm{KL}}(f_{\boldsymbol{\Theta}}; \boldsymbol{x})$ denote the KL regularizer applied to the output of the encoder $f_{\boldsymbol{\Theta}} \in \mathcal{F}_{\boldsymbol{\Xi}}$ given input $\boldsymbol{x}$. Then, with high probability, it holds:*

$$\sup_{f_{\boldsymbol{\Theta}} \in \mathcal{F}_{\boldsymbol{\Xi}}} \left[ \mathbb{E}_{\boldsymbol{x} \sim p(\boldsymbol{x})} D_{\mathrm{KL}}(f_{\boldsymbol{\Theta}}; \boldsymbol{x}) - \frac{1}{N} \sum_{i=1}^N D_{\mathrm{KL}}(f_{\boldsymbol{\Theta}}; \boldsymbol{x}_i) \right] \leq \tilde{\mathcal{O}}(1/\sqrt{N}). \tag{24}$$

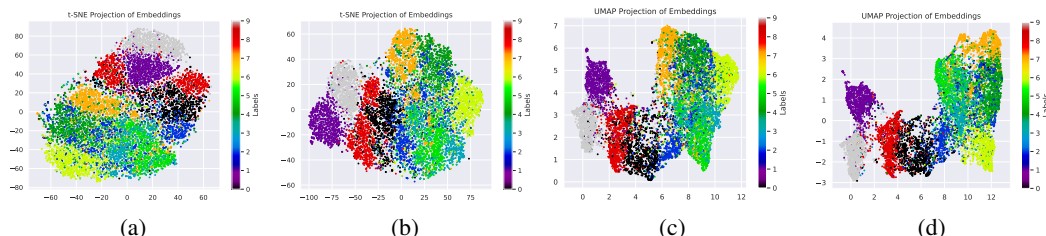

(a)          (b)          (c)          (d)

Figure 2: Embedding visualization for SimCLR and VSimCLR on CIFAR-10 test set. (a) t-SNE of SimCLR; (b) t-SNE of VSimCLR; (c) UMAP of SimCLR; (d) UMAP of VSimCLR. VSimCLR preserves the characteristic cluster structure of contrastive learning while introducing probabilistic embeddings regularized by (17).

Theorem 3.3 shows that the generalization gap of the KL term decays as $\tilde{\mathcal{O}}(1/\sqrt{N})$. In contrast, as shown in (Hieu et al., 2025, Theorem 1), the gap of InfoNCE scales as $\tilde{\mathcal{O}}(1)$ and does not improve with the number of negative samples $N$. Moreover, their analysis assumes that $\{x_i\}_{i=1}^N$ are drawn from class-conditional distributions, which is a strictly stronger assumption than our i.i.d. assumption from the marginal distribution. Therefore, the KL regularizer does not degrade the generalization guarantees of InfoNCE, while also providing principled uncertainty quantification.

## 4 EXPERIMENTS

We evaluate VCL with SimCLR and SupCon across five aspects: (i) embedding visualization, (ii) dimensional collapse, (iii) mutual information between embeddings and labels, (iv) classification accuracy, and (v) implications of distributional embeddings. Implementation and training details are provided in Appendix D.1.

### 4.1 EMBEDDING VISUALIZATION

Figure 2 presents t-SNE (Van der Maaten & Hinton, 2008) and UMAP (McInnes et al., 2018) projections of the embeddings learned by SimCLR and VSimCLR on the CIFAR-10 test set. Although VSimCLR incorporates an additional KL-regularizer, it preserves the characteristic cluster structure induced by contrastive learning. This confirms that our distributional embeddings retain the semantic information learned by contrastive methods.

### 4.2 DIMENSIONAL COLLAPSE

Contrastive learning methods such as SimCLR often suffer from *dimensional collapse*, where embeddings concentrate in a low-dimensional subspace, underutilizing the full capacity of the representation space (Jing et al., 2021). To quantify this effect, let $\{z_i\}_{i=1}^N$ be the test-set embeddings and their covariance matrix $C = \frac{1}{N}\sum_{i=1}^N (z_i - \bar{z})(z_i - \bar{z})^\top$, with $\bar{z} = \frac{1}{N}\sum_{i=1}^N z_i$. Figure 3 shows the singular values of $C$ for SimCLR and VSim-CLR. VSimCLR produces a substantially flatter spectrum, indicating a higher effective rank and thus mitigating dimensional collapse. Remarkably, on CIFAR-100, VSimCLR nearly doubles the number of dominant components compared to SimCLR. These results demonstrate that VSimCLR not only preserves semantic clustering but also leverages the embedding space more fully, and can be combined with other collapse-mitigation strategies for further gains. Additional experiments on Caltech-256 and Tiny-ImageNet (Figure 7, Appendix D.2) exhibit similar behavior.

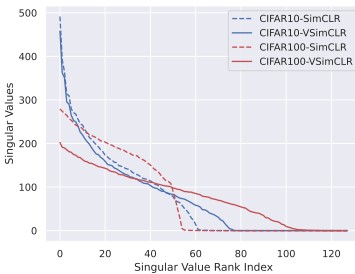

Figure 3: Singular-value spectrum of the embedding covariance on CIFAR-10 and CIFAR-100. VSim-CLR mitigates dimensional collapse.

Table 1: Classification accuracy on various datasets. We report top-1 and top-5 accuracies.

| METHOD | CIFAR-10 | | CIFAR-100 | | TINY-IMAGENET | | STL10 | | CALTECH256 | |
|--------|------|------|------|------|------|------|------|------|------|------|
| | Top1 | Top5 | Top1 | Top5 | Top1 | Top5 | Top1 | Top5 | Top1 | Top5 |
| SIMCLR | 78.42 | 98.52 | 49.56 | 78.84 | 38.95 | 66.89 | 60.44 | 95.80 | 43.14 | 66.15 |
| VSIMCLR | 81.48 | 98.95 | 54.58 | 82.87 | 37.70 | 66.06 | 60.11 | 92.00 | 48.50 | 69.99 |
| SUPCON | 93.60 | 99.71 | 70.79 | 89.11 | 57.60 | 77.16 | 75.88 | 98.51 | 87.06 | 91.64 |
| VSUPCON | 93.85 | 99.68 | 71.66 | 89.42 | 48.30 | 72.84 | 75.76 | 96.99 | 83.06 | 91.29 |

## 4.3 MUTUAL INFORMATION COMPARISON

Figure 4 reports the estimated mutual information $I(z; c)$ between the learned embeddings $z$ and their true class labels $c$ of CIFAR-10. We compute this using the Mixed KSG estimator (Gao et al., 2017), which is well-suited for mixed or multimodal distributions.

Both VSimCLR and VSupCon achieve mutual information on par with—or slightly exceeding—their non-variational counterparts. These results indicate that VSimCLR ultimately preserves—or even improves—information between embeddings and labels, while also producing rich distributional representations.

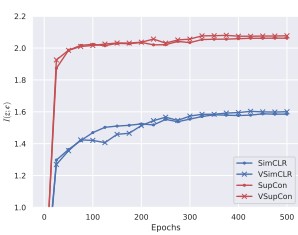

Figure 4: Estimate of $I(z; c)$.

## 4.4 CLASSIFICATION

For classification, we use the posterior mean $\mu_{\mathbf{x}}$ as the embedding and train a linear classifier.[2] Table 1 reports Top-1 and Top-5 accuracies on CIFAR-10, CIFAR-100, Tiny-ImageNet, STL-10, and Caltech-256. VSimCLR outperforms SimCLR on CIFAR-10 and CIFAR-100 in Top-1 accuracy, with similar gains in Top-5. On Caltech-256, VSimCLR also improves Top-1 accuracy substantially. Performance on Tiny-ImageNet and STL-10 remains comparable, with slight decreases (within experimental variance) likely due to the KL regularizer.

SupCon provides supervised baselines, and VSupCon further improves Top-1 accuracy on CIFAR-10 and CIFAR-100. Modest declines on Tiny-ImageNet, STL-10, and Caltech-256 reflect the trade-off of adding the KL term on datasets with higher complexity or fewer samples. We hypothesize that the drop in VSupCon arises from two factors: (i) VSimCLR's objective coincides with the VCL objective in (19), whereas VSupCon's does not, creating a mismatch that may hinder proper ELBO maximization; and (ii) SupCon directly optimizes embeddings for classification, so an added KL term can conflict with that objective. We further study the effect of the KL regularizer in Appendix D.4.

Although VCL is not designed to boost classification accuracy, VSimCLR consistently match or exceed their deterministic counterparts. This demonstrates that probabilistic embeddings preserve the alignment and uniformity (Wang & Isola, 2020), while yielding meaningful uncertainty proxy.

## 4.5 IMPLICATIONS OF DISTRIBUTIONAL EMBEDDINGS

**Examples of CIFAR-10 with posterior.** We illustrate the interpretability of posterior using examples from CIFAR-10. Figure 8 displays sample images alongside the log-determinant $\log \det(K)$ of their posterior covariance $K$ learned by VSimCLR. Top-row images are common class members and exhibit larger $\log \det(K)$—indicating broader posterior dispersion—whereas bottom-row images are atypical or uncommon with smaller $\log \det(K)$, reflecting more concentrated posteriors.[3]

---

[2]In Appendix D.3, we present additional results exploring various VSimCLR design choices.

[3]$\log \det K$ quantifies the *dispersion of the posterior in embedding space*, which reflects *typicality* rather than label uncertainty. Larger values correspond to more "typical" samples with many latent realizations consistent with the data manifold, whereas smaller values indicate more "unique" or outlier samples with tightly concentrated posteriors. A generative analogy may help understanding: if an outlier image had an extremely large posterior variance, then samples drawn from the prior would reproduce that outlier far too often—contradicting its rarity. Hence, larger variance corresponds to "typical" not "uncertain" inputs.

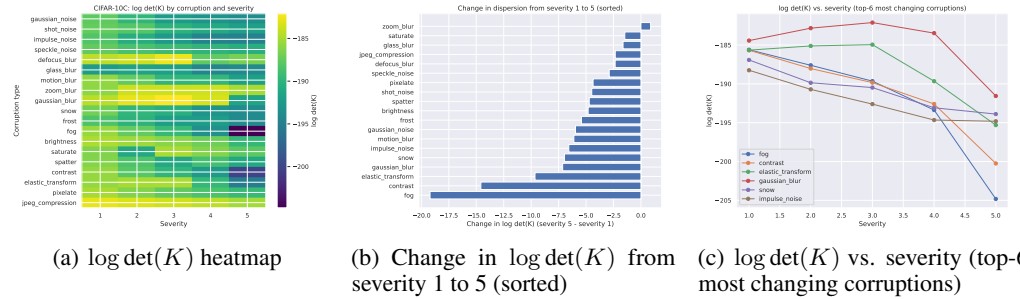

(a) $\log \det(K)$ heatmap

(b) Change in $\log \det(K)$ from severity 1 to 5 (sorted)

(c) $\log \det(K)$ vs. severity (top-6 most changing corruptions)

Figure 5: $\log \det(K)$ of **VSimCLR** embeddings on CIFAR-10C under different corruption types and severities. "Severity" denotes the corruption level. Exact $\log \det(K)$ values are in Table 7.

**Relationship between label-entropy and $\log \det(K)$.** We quantify the relationship between posterior covariance and uncertainty using CIFAR-10H (Peterson et al., 2019) and CIFAR-10C (Hendrycks & Dietterich, 2019). Figure 9 and 10 plot posterior dispersion against the entropy of the CIFAR-10H soft labels (Ishida et al., 2023; Jeong et al., 2023); the negative slope of the linear fit (red line) indicates that images with lower $\log \det(K)$—i.e., more concentrated posteriors—tend to have higher label entropy and thus greater ambiguity. Next, using CIFAR-10C, we examine how posterior covariance varies with corruption severity, which correlates with label uncertainty. Figures 5 and 11 show that $\log \det(K)$ decreases as corruption strength increases, implying that lower posterior dispersion corresponds to higher uncertainty, consistent with Figure 9. These results demonstrate that the dispersion of the learned posterior correlates with semantic uncertainty, highlighting the practical interpretability of VCL's distributional embeddings.

**Use case of posterior.** As an example application of posterior, we consider CIFAR-100 under a label-scarce setting in which only a small number of labels per class are available to train a linear classifier. Table 6 reports accuracies for SimCLR, VSimCLR, and VSimCLR+wt, with classifiers trained using cross-entropy (CE). Here, "+wt" denotes a weighted CE in which sample weights are proportional to posterior covariance to downweight ambiguous examples. Specifically, we use $\mathcal{L}_{wCE} = \sum_{i=1}^{N} w_i \log \phi_{c_i}(z_i)$, with $w_i \propto \log \det(K)$ (after normalization), where $\phi_{c_i}(z_i)$ is the estimated probability of the true class. Table 6 shows that VCL variants improve over SimCLR and SupCon, with smaller gains for SupCon since it already leverages labels during pretraining. Moreover, weighting by posterior covariance further improves performance, supporting probabilistic embeddings as a confidence proxy. Additional experiments and discussion are provided in Appendix D.5.

This counterintuitive finding—that typical (i.e., common) samples exhibit larger posterior dispersion—parallels the observation in concurrent work by Guth et al. (2025), albeit under different settings: (i) Quantity: we analyze latent-space posterior via $\log \det K$, whereas they study input-space marginal density $p(x)$; (ii) Observation: typical samples have larger $\log \det K$, while they have lower marginal density. Although the quantities are measured in different spaces, both results indicate that typical samples are not the highest-density points. In our case, typical images yield larger posterior dispersion and atypical images smaller dispersion; since dispersion is inversely related to peak density, our result aligns with (Guth et al., 2025). Hence, in both settings, "typical" $\neq$ "highest-density."

## 5 CONCLUSION

We introduced *Variational Contrastive Learning* (VCL), a decoder-free ELBO framework that equips contrastive learning with probabilistic embeddings by treating InfoNCE as a surrogate reconstruction term and adding a KL penalty to a uniform spherical prior. Instantiated as VSimCLR and VSup-Con—sampling from $q_\theta(z \mid x)$ with a normalized KL—VCL preserves the strengths of contrastive embeddings, mitigates dimensional collapse, maintains or improves label mutual information, and matches or surpasses deterministic baselines, while yielding calibrated posterior uncertainty. Analysis of posterior-covariance dispersion further shows a consistent pattern—also noted in concurrent diffusion-model work (Guth et al., 2025)—where typical samples exhibit larger covariance and atypical/outlier samples show smaller dispersion.

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

# A    RELATED WORK

## A.1    CONTRASTIVE LEARNING

Self-supervised contrastive learning methods (Chen et al., 2020; Tian et al., 2020) train an encoder $f \colon \mathcal{X} \to \mathcal{S}^{d_z - 1}$ by drawing semantically related views (positives) together in the embedding space while pushing unrelated views (negatives) apart. In the standard setup, each example is treated as its own category, and only its augmented copies count as positives. A variety of contrastive objectives—such as InfoNCE (Oord et al., 2018), Debiased Contrastive Loss (Chuang et al., 2020), Unbiased Contrastive Loss (Barbano et al., 2022), triplet-based losses (Chopra et al., 2005; Hermans et al., 2017), and others—have been used to learn robust representations for tasks ranging from dense prediction in computer vision (Wang et al., 2021) to multimodal alignment (Radford et al., 2021; Girdhar et al., 2023; Jeong et al., 2024b). InfoNCE (Oord et al., 2018) in particular has been shown to lower-bound mutual information (Poole et al., 2019), and subsequent work has revealed that its empirical success hinges on a balance of *alignment* and *uniformity* in the learned embeddings (Tschannen et al., 2019; Wang & Isola, 2020). In the supervised setting, SupCon (Khosla et al., 2020) extends this idea by using class labels to define positive pairs among same-class samples, often surpassing cross-entropy training in downstream performance. ProjNCE, a generalization of SupCon (Jeong & Hero, 2025), modifies SupCon loss so that it becomes a proper mutual information lower bound.

## A.2    PROBABILISTIC CONTRASTIVE LEARNING

A growing body of work has begun to integrate probabilistic latent-variable modeling with contrastive objectives. In the video domain, Park et al. represent each video clip as a Gaussian and combine them into a mixture model, learning these distributions via a stochastic contrastive loss that captures clip-level uncertainty and obviates complex augmentation schemes (Park et al., 2022). For 3D point clouds, Wang et al. propose a Generative Variational-Contrastive framework that models latent features as Gaussians, enforces distributional consistency across positive pairs by combining the variational autoencoder and contrastive learning (Wang et al., 2024a). In graph representation learning, Xie and Giraldo introduce Subgraph Gaussian Embedding Contrast, which maps subgraphs into a structured Gaussian space and employs optimal-transport distances for robust contrastive objectives, yielding improved classification and link-prediction performance (Xie & Giraldo, 2024).

On the theoretical front, Zimmermann et al. prove that contrastive objectives invert the data-generating process under mild conditions, uncovering a deep connection to nonlinear independent component analysis (Zimmermann et al., 2021). With a more generalized setting, Kirchhof et al. extend the InfoNCE loss so that the encoder predicts a full posterior distribution rather than a point, and prove that these distributions asymptotically recover the true aleatoric uncertainty of the data-generating process (Kirchhof et al., 2023).

## A.3    VARIATIONAL INFERENCE AND CONTRASTIVE LEARNING

The most closely related line of work frames contrastive learning within a latent-variable inference paradigm via Recognition-Parametrised Models (RPMs) (Aitchison & Ganev, 2021; Walker et al., 2023). Aitchison and Ganev introduce RPMs as a class of Bayesian models whose (unnormalized) likelihood is defined implicitly through a recognition network (Aitchison & Ganev, 2021). They show that, under RPMs, the ELBO decomposes into mutual information minus a KL term (up to a constant), and that for a suitable choice of prior the infinite-sample InfoNCE objective coincides with this ELBO. Walker et al. consider RPMs by assuming conditional independence of observations given latent variables, and develop an EM algorithm that achieves exact maximum-likelihood learning for discrete latents along with principled posterior inference (Walker et al., 2023).

Other works recast variational inference itself as a contrastive estimation task. Rhodes and Gutmann's Variational Noise-Contrastive Estimation (VNCE) derives a variational lower bound to the standard NCE objective, enabling joint learning of model parameters and latent posteriors in unnormalized models (Rhodes & Gutmann, 2019). More recently, Ward et al. propose SoftCVI, which treats VI as a classification problem: they generate "soft" pseudo-labels from the unnormalized posterior and optimize a contrastive-style objective that yields zero-variance gradients at the optimum (Ward et al., 2025).

### A.4 DIMENSIONAL COLLAPSE

In contrastive self-supervised learning, several approaches have been proposed to prevent dimensional collapse by regularizing either the embedding projector or the second-order statistics of the representations. Jing *et al.* (Jing et al., 2021) first demonstrated that, despite the repulsive effect of negative samples, embeddings can still collapse to a low-dimensional subspace due to a combination of strong augmentations and implicit low-rank bias in weight updates. They introduced DirectCLR, which fixes a low-rank diagonal projector during training; this projector enforces the embeddings to occupy a predetermined subspace and was shown empirically to outperform SimCLR's learned linear projector.

Following this, several works have designed novel loss functions that explicitly regularize the covariance or cross-correlation of the embedding vectors. Ermolov *et al.* (Ermolov et al., 2021) apply a whitening MSE loss so that positive pairs match under mean-square error while enforcing identity covariance. Barlow Twins (Zbontar et al., 2021) minimize the deviation of the normalized cross-correlation matrix from the identity, effectively performing "soft whitening" to reduce redundancy. VICReg (Bardes et al., 2021) further augments this idea by combining variance, invariance, and covariance regularizers to avoid collapse without using negative samples; notably, VICReg allows its two branches to use different architectures or even modalities, enabling joint embedding across data types. More recently, He *et al.* (He et al., 2024) showed that orthogonal regularization of encoder weight matrices preserves representation diversity and prevents collapse.

## B PROOFS

### B.1 PROOF OF LEMMA 3.1

*Proof.* With any auxiliary probability function $r(\boldsymbol{z}'|\boldsymbol{x})$ and Jensen's inequality, we have

$$\mathbb{E}_{q(\boldsymbol{z}|\boldsymbol{x})}[\log p(\boldsymbol{x}|\boldsymbol{z})] \geq \mathbb{E}_{q(\boldsymbol{z}|\boldsymbol{x})r(\boldsymbol{z}'|\boldsymbol{x})}\left[\log \frac{p(\boldsymbol{z}'|\boldsymbol{x})p(\boldsymbol{x}|\boldsymbol{z}')}{r(\boldsymbol{z}'|\boldsymbol{x})}\right]$$

$$\overset{(a)}{=} \mathbb{E}_{q(\boldsymbol{z}|\boldsymbol{x})r(\boldsymbol{z}'|\boldsymbol{x})}[\log p(\boldsymbol{z}'|\boldsymbol{z})] + \mathbb{E}_{r(\boldsymbol{z}'|\boldsymbol{x})}[\log p(\boldsymbol{x}|\boldsymbol{z}')] + H(r(\boldsymbol{z}'|\boldsymbol{x}))$$

$$= \mathbb{E}_{q(\boldsymbol{z}|\boldsymbol{x})q(\boldsymbol{z}'|\boldsymbol{x})}[\log p(\boldsymbol{z}'|\boldsymbol{z})] + \text{const.}, \tag{25}$$

where (a) follows by choosing $r(z'|x) = q(z'|x)$. This proves Lemma 3.1. $\qquad\square$

### B.2 PROOF OF PROPOSITION 3.2

*Proof.* Optimal critic (Ma & Collins, 2018) for InfoNCE satisfies that

$$\psi^{\star}(\boldsymbol{x}, \boldsymbol{z}) \propto \log \frac{p(\boldsymbol{x}|\boldsymbol{z})}{p(\boldsymbol{x})} + \alpha(\boldsymbol{z}), \tag{26}$$

where $\alpha(\boldsymbol{z})$ only depends on $\boldsymbol{z}$. With the optimal critic, we then have

$$I_{\text{NCE}}(\boldsymbol{x}; \boldsymbol{x}') = -\mathbb{E}\left[\log \frac{e^{\psi(\boldsymbol{z}, \boldsymbol{z}'_i)}}{\sum_{j=1}^{N} e^{\psi(\boldsymbol{z}, \boldsymbol{z}'_j)}}\right]$$

$$= -\mathbb{E}\left[\log \frac{p(\boldsymbol{z}|\boldsymbol{z}'_i)}{\sum_{j=1}^{N} p(\boldsymbol{z}|\boldsymbol{z}'_j)}\right]$$

$$= -\mathbb{E}\left[\log \frac{p(\boldsymbol{z}|\boldsymbol{z}'_i)}{\frac{1}{N}\sum_{j=1}^{N} p(\boldsymbol{z}|\boldsymbol{z}'_j)}\right] + \log N. \tag{27}$$

Given $\boldsymbol{z}$, since $p(\boldsymbol{z}|\boldsymbol{z}'_j)$, $j \in \{1, 2, \cdots, N\}$ are i.i.d. with $\mathbb{E}[p(\boldsymbol{z}|\boldsymbol{z}'_j)] = p(\boldsymbol{z}) < \infty$, the strong law of large numbers yields

$$\lim_{N \to \infty} \frac{1}{N} \sum_{j=1}^{N} p(\boldsymbol{z}|\boldsymbol{z}'_j) = p(\boldsymbol{z}). \tag{28}$$

The continuous mapping theorem then gives

$$\lim_{N \to \infty} \log \frac{p(\boldsymbol{z}|\boldsymbol{z}'_i)}{\frac{1}{N}\sum_{j=1}^{N} p(\boldsymbol{z}|\boldsymbol{z}'_j)} = \log \frac{p(\boldsymbol{z}|\boldsymbol{z}'_i)}{p(\boldsymbol{z})}. \tag{29}$$

Rearranging (26) and taking $N \to \infty$, we obtain

$$\lim_{N \to \infty} \{I_{\mathrm{NCE}}(\boldsymbol{x}; \boldsymbol{x}') + \log N\} = \lim_{N \to \infty} \mathbb{E}\left[\log \frac{p(\boldsymbol{z}|\boldsymbol{z}'_i)}{\frac{1}{N}\sum_{j=1}^{N} p(\boldsymbol{z}|\boldsymbol{z}'_j)}\right]$$

$$\stackrel{\text{(a)}}{=} \mathbb{E}\left[\lim_{N \to \infty} \log \frac{p(\boldsymbol{z}|\boldsymbol{z}'_i)}{\frac{1}{N}\sum_{j=1}^{N} p(\boldsymbol{z}|\boldsymbol{z}'_j)}\right]$$

$$= \mathbb{E}\left[\log \frac{p(\boldsymbol{z}|\boldsymbol{z}'_i)}{p(\boldsymbol{z})}\right], \tag{30}$$

where the equality (a) follows by dominated convergence theorem that is verifiable using the fact that

$$\mathbb{E}\left[\log \frac{p(\boldsymbol{z}|\boldsymbol{z}'_i)}{\frac{1}{N}\sum_{j=1}^{N} p(\boldsymbol{z}|\boldsymbol{z}'_j)}\right] = \mathbb{E}\left[\log p(\boldsymbol{z}|\boldsymbol{z}'_i) - \log \frac{1}{N}\sum_{j=1}^{N} p(\boldsymbol{z}|\boldsymbol{z}'_j)\right]$$

$$\leq \mathbb{E}\left[\log g(\boldsymbol{z}) - \log \epsilon\right]$$

$$\leq \log \mathbb{E}\left[g(\boldsymbol{z})\right] - \log \epsilon$$

$$< \infty. \tag{31}$$

Rewriting (30) gives

$$\lim_{N \to \infty} \{I_{\mathrm{NCE}}(\boldsymbol{x}; \boldsymbol{x}') + \log N\}$$

$$= \mathbb{E}\left[\log \frac{p(\boldsymbol{z}|\boldsymbol{z}'_i)}{p(\boldsymbol{z})}\right]$$

$$= \mathbb{E}\left[\log \frac{p(\boldsymbol{z}'_i|\boldsymbol{z})}{p(\boldsymbol{z}'_i)}\right]$$

$$= \mathbb{E}_{q_\theta(\boldsymbol{z}'_i|\boldsymbol{x})q_\theta(\boldsymbol{z}|\boldsymbol{x})}\left[\log p(\boldsymbol{z}'_i|\boldsymbol{z})\right] + \mathbb{E}_{q_\theta(\boldsymbol{z}'_i|\boldsymbol{x})}\left[\log p(\boldsymbol{z}'_i)\right]$$

$$= \mathbb{E}_{q_\theta(\boldsymbol{z}'_i|\boldsymbol{x})q_\theta(\boldsymbol{z}|\boldsymbol{x})}\left[\log p(\boldsymbol{z}'_i|\boldsymbol{z})\right] + \mathbb{E}_{q_\theta(\boldsymbol{z}'_i|\boldsymbol{x})}\left[\log \frac{p(\boldsymbol{z}'_i)}{q_\theta(\boldsymbol{z}'_i|\boldsymbol{x})}\right] + \mathbb{E}_{q_\theta(\boldsymbol{z}'_i|\boldsymbol{x})}\left[\log q_\theta(\boldsymbol{z}'_i|\boldsymbol{x})\right]$$

$$= \mathbb{E}_{q_\theta(\boldsymbol{z}'_i|\boldsymbol{x})q_\theta(\boldsymbol{z}|\boldsymbol{x})}\left[\log p(\boldsymbol{z}'_i|\boldsymbol{z})\right] - D(q_\theta(\boldsymbol{z}'_i|\boldsymbol{x})\|p(\boldsymbol{z}'_i)) - H(q_\theta(\boldsymbol{z}'_i|\boldsymbol{x})). \tag{32}$$

Substituting $\boldsymbol{z}'_i$ into $\boldsymbol{z}'$, this concludes the proof of Proposition 3.2 $\qquad\square$

### B.3 Formal statement of Theorem 3.3 and its proof

In this section, we provide detailed problem setup, formal statement of Theorem 3.3, its extension to random augmentation, and the proof of Theorem 3.3.

#### B.3.1 Problem setup and formal statement of Theorem 3.3

First, we introduce the class of the deep forward neural network with bounded weights and Lipschitz activations.

Let $\{\boldsymbol{x}_i\}_{i=1}^{N} \subset \mathcal{X}$ be independent and identically distributed samples from a distribution $\mathcal{D}$ on $\mathcal{X}$. We consider depth-$L$ feed-forward networks $f_{\boldsymbol{\Theta}} : \mathcal{X} \to \mathbb{R}^{2d}$ of the form

$$f_{\boldsymbol{\Theta}}(\boldsymbol{x}) = \alpha_L\Big(\boldsymbol{\Theta}^L \, \alpha_{L-1}\big(\boldsymbol{\Theta}^{L-1} \cdots \alpha_1(\boldsymbol{\Theta}^1 \boldsymbol{x})\big)\Big),$$

where each activation $\alpha_\ell : \mathbb{R}^{d_\ell} \to \mathbb{R}^{d_\ell}$ is $\sigma_\ell$-Lipschitz. For each layer $\ell = 1, \ldots, L$, the weight matrices lie in

$$\boldsymbol{\Xi}_\ell := \big\{\boldsymbol{\Theta}^\ell \in \mathbb{R}^{d_\ell \times d_{\ell-1}} : \|\boldsymbol{\Theta}^\ell\|_2 \leq \rho_\ell, \ \|(\boldsymbol{\Theta}^\ell - \boldsymbol{\Theta}_0^\ell)^\top\|_{2,1} \leq a_\ell\big\}, \tag{33}$$

where $\|\cdot\|_2$ is the spectral norm and $\|\cdot\|_{2,1}$ is the sum of column-wise $\ell_2$ norms, and $\{\boldsymbol{\Theta}_0^\ell\}_{\ell=1}^L$ are fixed reference matrices. Set $\boldsymbol{\Xi} = \prod_{\ell=1}^L \boldsymbol{\Xi}_\ell$ and define

$$\mathcal{F}_{\boldsymbol{\Xi}} = \{f_{\boldsymbol{\Theta}} : \boldsymbol{\Theta} \in \boldsymbol{\Xi}\}. \tag{34}$$

In VCL the encoder output splits as

$$f_{\boldsymbol{\Theta}}(\boldsymbol{x}) = \begin{pmatrix} \mu_{\boldsymbol{\Theta}}(\boldsymbol{x}) \\ \varepsilon_{\boldsymbol{\Theta}}(\boldsymbol{x}) \end{pmatrix}, \quad \mu_{\boldsymbol{\Theta}}, \varepsilon_{\boldsymbol{\Theta}} : \mathcal{X} \to \mathbb{R}^d. \tag{35}$$

Given $\boldsymbol{x}$, the KL regularizer by output of $f_{\boldsymbol{\Theta}}(\boldsymbol{x})$ in (35) is

$$D(f_{\boldsymbol{\Theta}}; \boldsymbol{x}) = \frac{1}{2d}\Big(\|\mu_{\boldsymbol{\Theta}}(\boldsymbol{x})\|_2^2 + \big\|\exp(\varepsilon_{\boldsymbol{\Theta}}(\boldsymbol{x}))\big\|_1 - d - \big\langle \mathbf{1}_d, \varepsilon_{\boldsymbol{\Theta}}(\boldsymbol{x})\big\rangle\Big). \tag{36}$$

**Theorem B.1.** *Fix $\delta \in (0,1)$. Under the assumptions on $\{\boldsymbol{x}_i\}_{i=1}^N$ and definitions on $\mathcal{F}_{\boldsymbol{\Xi}}$ (34) and $D(f_{\boldsymbol{\Theta}}; \boldsymbol{x})$ (36), it holds that*

$$\sup_{f_{\boldsymbol{\Theta}} \in \mathcal{F}_{\boldsymbol{\Xi}}} \Big[ \mathbb{E}_{\boldsymbol{x} \sim \mathcal{D}} \, D(f_{\boldsymbol{\Theta}}; \boldsymbol{x}) - \frac{1}{N} \sum_{i=1}^N D(f_{\boldsymbol{\Theta}}; \boldsymbol{x}_i) \Big]$$

$$\leq \widetilde{\mathcal{O}}\Big( \frac{\eta \, B_x \prod_{\ell=1}^L (\rho_\ell \, \sigma_\ell)}{\sqrt{N}} \Big[ \sqrt{\log W} \, \Big( \sum_{\ell=1}^L \big(a_\ell/\rho_\ell\big)^{2/3} \Big)^{3/2} \vee \sqrt{\log(1/\delta)} \Big] \Big),$$

*with probability at least $1 - \delta$, where*

$$W = \max_{\ell \in [L]} d_\ell, \quad B_x = \sup_{\boldsymbol{x} \in \mathcal{X}} \|\boldsymbol{x}\|_2, \quad \eta = \mathcal{O}\Big(1 \vee e^{B_x \prod_{\ell=1}^L \rho_\ell \, \sigma_\ell}\Big).$$

**Random augmentations.** One might be concerned that the encoder in (35) uses raw inputs, whereas in practice each sample $\boldsymbol{x}_i$ is fed through two random augmentations $t_i', t_i'' \sim \mathcal{T}$ (see Section 2.1). To deal with this concern, we show that exactly the same bound of Theorem B.1 holds under this more practical setting.

Let $\mathcal{T}$ be a distribution over augmentation maps $t : \mathcal{X} \to \mathcal{X}$. Define the *augmented data law*

$$\mathcal{D}_{\mathrm{aug}} := \mathrm{Law}_{\boldsymbol{x} \sim \mathcal{D}, \, t \sim \mathcal{T}}\big(t(\boldsymbol{x})\big) = \mathcal{T}_\# \mathcal{D},$$

where $\mathcal{T}_\# \mathcal{D}$ denotes the push-forward of the measure $\mathcal{D}$ through the (random) map distribution $\mathcal{T}$.

If

$$t_i \overset{\mathrm{i.i.d.}}{\sim} \mathcal{T}, \quad \boldsymbol{x}_i' = t_i(\boldsymbol{x}_i),$$

then by construction

$$\boldsymbol{x}_1', \ldots, \boldsymbol{x}_N' \overset{\mathrm{i.i.d.}}{\sim} \mathcal{D}_{\mathrm{aug}},$$

so the conditions of Theorem B.1 remain satisfied with $\mathcal{D}_{\mathrm{aug}}$ in place of $\mathcal{D}$.

**Corollary B.2.** *Let $\{t_i'\}_{i=1}^N$ and $\{t_i''\}_{i=1}^N$ be independent samples from $\mathcal{T}$, and set $\boldsymbol{x}_i' = t_i'(\boldsymbol{x}_i)$, $\boldsymbol{x}_i'' = t_i''(\boldsymbol{x}_i)$. Under the assumptions of Theorem B.1, with probability at least $1 - \delta$ we have*

$$\sup_{f_{\boldsymbol{\Theta}} \in \mathcal{F}_{\boldsymbol{\Xi}}} \Big[ \mathbb{E}_{\boldsymbol{x} \sim \mathcal{D}, \, t \sim \mathcal{T}} \, D\big(f_{\boldsymbol{\Theta}}; t(\boldsymbol{x})\big) - \frac{1}{2N} \sum_{i=1}^N \big(D(f_{\boldsymbol{\Theta}}; \boldsymbol{x}_i') + D(f_{\boldsymbol{\Theta}}; \boldsymbol{x}_i'')\big) \Big]$$

$$\leq \widetilde{\mathcal{O}}\Big( \frac{\eta \, B_{x,t} \prod_{\ell=1}^L (\rho_\ell \sigma_\ell)}{\sqrt{N}} \Big[ \sqrt{\log W} \, \Big( \sum_{\ell=1}^L (a_\ell/\rho_\ell)^{2/3} \Big)^{3/2} \vee \sqrt{\log(1/\delta)} \Big] \Big),$$

*where $B_{x,t} = \sup_{\boldsymbol{x} \in \mathcal{X}, \, t \in \mathcal{T}} \|t(\boldsymbol{x})\|_2$.*

One might wonder whether $B_{x,t}$ is finite in Corollary B.2. In practice, common vision augmentations—such as random resized cropping and horizontal flips (Krizhevsky et al., 2012; Perez & Wang, 2017), color jittering (brightness, contrast, saturation perturbations) (Shorten & Khoshgoftaar, 2019; Chen et al., 2020), and additive Gaussian blur or noise (Hendrycks et al., 2020)—always produce

outputs that remain within a bounded neighborhood of the original inputs. For example, cropping a $256 \times 256$ image to $224 \times 224$ keeps all pixel values in their original range, and color jitter is applied with limited intensity so that augmented images lie close in Euclidean norm to the source. Even when combining multiple operations (as in AugMix), the convex mixtures ensure augmented samples do not "drift" outside the natural image manifold. Hence, $\sup_{\boldsymbol{x} \in \mathcal{X}, \, t \in \mathcal{T}} \|t(\boldsymbol{x})\|_2$ remains finite under these widely adopted schemes.

Before proving Theorem B.1, we need to introduce some background on Rademacher complexity, Dudley's entropy-integral bound, and Lipschitz of KL divergence, which are necessary in the proof.

### B.3.2 BACKROUND: RADEMACHER COMPLEXITY AND DUDLEY'S ENTROPY-INTEGRAL BOUND

We introduce the background on Rademacher complexity, Dudley's entropy-integral bound, and covering number for the function class $\mathcal{F}_{\boldsymbol{\Xi}}$ in (34), which are used in the proof of Theorem B.1.

**Lemma B.3** (Rademacher generalization bound). *(Mohri et al., 2012) Let $\mathcal{X}$ be a vector space and $\mathcal{D}$ a distribution over $\mathcal{X}$. Let $\mathcal{F}$ be a class of functions $f : \mathcal{X} \to [a, b]$. Draw samples $\mathcal{S} = \{\boldsymbol{x}_i\}_{i=1}^N \overset{\text{i.i.d}}{\sim} \mathcal{D}^n$, and define the empirical Rademacher complexity*

$$\hat{\mathfrak{R}}_S(\mathcal{F}) \;=\; \mathbb{E}_\sigma \left[ \sup_{f \in \mathcal{F}} \frac{1}{N} \sum_{i=1}^N t_i \, f(\boldsymbol{x}_i) \right], \tag{37}$$

*where $t_i \in \{-1, 1\}$ are independent Rademacher variables. Then for any $\delta \in (0, 1)$, with probability at least $1 - \delta$,*

$$\sup_{f \in \mathcal{F}} \left| \mathbb{E}_{\boldsymbol{x} \sim \mathcal{D}}[\, f(\boldsymbol{x})\,] \;-\; \frac{1}{N} \sum_{i=1}^N f(\boldsymbol{x}_i) \right| \;\leq\; 2\,\hat{\mathfrak{R}}_S(\mathcal{F}) \;+\; 3\,(b - a) \sqrt{\frac{\ln(2/\delta)}{2N}}.$$

**Lemma B.4** (Dudley's entropy-integral bound). *(Bartlett et al., 2017, Lemma 8.5) Let $\mathcal{X}$ be a vector space and $\mathcal{F}$ a class of functions $f : \mathcal{X} \to \mathbb{R}$. For a sample $\mathcal{S} = \{\boldsymbol{x}_i\}_{i=1}^N$, define*

$$\|f\|_{L_2(\mathcal{S})} \;=\; \left( \frac{1}{N} \sum_{i=1}^N f(\boldsymbol{x}_i)^2 \right)^{1/2}, \quad B_\mathcal{F} \;=\; \sup_{f \in \mathcal{F}} \|f\|_{L_2(\mathcal{S})}.$$

*Then the empirical Rademacher complexity of $\mathcal{F}$ (37) satisfies*

$$\hat{\mathfrak{R}}_{\mathcal{S}}(\mathcal{F}) \;\leq\; \inf_{\alpha > 0} \left[ 4\,\alpha \;+\; \frac{12}{\sqrt{N}} \int_\alpha^{B_\mathcal{F}} \sqrt{\ln \mathcal{N}\big(\mathcal{F}, \varepsilon, L_2(\mathcal{S})\big)} \, d\varepsilon \right],$$

*where $\mathcal{N}(\mathcal{F}, \varepsilon, L_2(\mathcal{S}))$ is the covering number of $\mathcal{F}$ under the $\| \cdot \|_{L_2(\mathcal{S})}$ metric.*

**Lemma B.5** (Covering-number bound for $\mathcal{F}_{\boldsymbol{\Xi}}$). *(Hieu et al., 2025, Proposition 4) Let $\mathcal{F}_{\boldsymbol{\Xi}}$ be the class of $L$-layer neural networks with bounded weights and Lipschitz activation functions, in (34). Let*

$$\mathcal{S} = \{\boldsymbol{x}_i\}_{i=1}^N \subset \mathcal{X}$$

*be any fixed dataset. Define*

$$\overline{\mathcal{R}}_{\boldsymbol{\Xi}}^{2/3} = \sum_{l=1}^L \big(a_l \, B_{l-1} \, \rho_{l+}\big)^{2/3},$$

$$\rho_{l+} = \rho_l \prod_{m=l+1}^L \big(\rho_m \, s_m\big),$$

$$B_l = \sup_{\boldsymbol{x} \in \mathcal{X}} \sup_{\boldsymbol{\Theta} \in \boldsymbol{\Xi}} \big\|f_{\boldsymbol{\Theta}}^{1 \to l}(\boldsymbol{x})\big\|_2,$$

*where $f_{\boldsymbol{\Theta}}^{1 \to l}(\boldsymbol{x})$ is the output of the first $l$ layers of the network. Then for every $\varepsilon > 0$,*

$$\log \mathcal{N}\big(\mathcal{F}_{\boldsymbol{\Xi}}, \varepsilon, L_{\infty, 2}(S)\big) \;\leq\; \frac{64\,\overline{\mathcal{R}}_{\boldsymbol{\Xi}}^2}{\varepsilon^2} \, \log\!\Big(\big(\tfrac{11\overline{\mathcal{R}}_{\boldsymbol{\Xi}}}{\varepsilon} + 7\big) \, N\,W\Big).$$

### B.3.3    $\ell_2$-LIPSCHITZNESS OF KL DIVERGENCE

We introduce the Lipschitzness of the KL divergence term $D(f_{\boldsymbol{\Theta}}; \boldsymbol{x})$ in (36) with respect to the $\ell_2$ norm when we assume the output of the encoder is bounded by $B_L$. The Lipschitzness of the KL divergence term is crucial for the proof of Theorem B.1. We will specify the bound $B_L$ later in the proof of Theorem B.1.

**Lemma B.6** ($\ell_2$–Lipschitzness of the KL term). *For some constant $B_L > 0$, define*

$$V_{B_L} := \left\{ [\boldsymbol{\mu}; \boldsymbol{\varepsilon}] \in \mathbb{R}^{2d} : \boldsymbol{\mu} \in \mathbb{R}^d, \boldsymbol{\varepsilon} \in \mathbb{R}^d, \|[\boldsymbol{\mu}; \boldsymbol{\varepsilon}]\|_2 \leq B_L. \right\},$$

*and let*

$$D([\boldsymbol{\mu}; \boldsymbol{\varepsilon}]) := \frac{1}{2d} \sum_{j=1}^{d} \left( \boldsymbol{\mu}_j^2 + \exp(\boldsymbol{\varepsilon}_j) - 1 - \boldsymbol{\varepsilon}_j \right). \tag{38}$$

*Then, for any two pairs $[\boldsymbol{\mu}; \boldsymbol{\varepsilon}], [\boldsymbol{\mu}'; \boldsymbol{\varepsilon}'] \in V_{B_L}$, it holds that*

$$\left| D([\boldsymbol{\mu}; \boldsymbol{\varepsilon}]) - D([\boldsymbol{\mu}'; \boldsymbol{\varepsilon}']) \right| \leq \frac{\eta}{\sqrt{d}} \cdot \left\| ([\boldsymbol{\mu}; \boldsymbol{\varepsilon}]) - ([\boldsymbol{\mu}'; \boldsymbol{\varepsilon}']) \right\|_2,$$

*where $\eta = B_L + (1 + \exp(B_L))/2$.*

*Proof.* Compute the gradient of $D([\boldsymbol{\mu}; \boldsymbol{\varepsilon}])$ in (38) as

$$\nabla D([\boldsymbol{\mu}; \boldsymbol{\varepsilon}]) = \frac{1}{d} \begin{bmatrix} \boldsymbol{\mu} \\ \frac{\exp(\boldsymbol{\varepsilon}) - \mathbf{1}_d}{2} \end{bmatrix} \tag{39}$$

where $\mathbf{1}_d := [1, \ldots, 1] \in \mathbb{R}^d$.

Fix $[\boldsymbol{\mu}; \boldsymbol{\varepsilon}], [\boldsymbol{\mu}'; \boldsymbol{\varepsilon}'] \in V_{B_L}$ arbitrarily. Then, applying Cauchy–Schwarz inequality with the convexity of $D([\boldsymbol{\mu}; \boldsymbol{\varepsilon}])$ yields

$$\left| D([\boldsymbol{\mu}; \boldsymbol{\varepsilon}]) - D([\boldsymbol{\mu}'; \boldsymbol{\varepsilon}']) \right| \leq \nabla D([\boldsymbol{\mu}; \boldsymbol{\varepsilon}])^\top ([\boldsymbol{\mu}; \boldsymbol{\varepsilon}] - [\boldsymbol{\mu}'; \boldsymbol{\varepsilon}']) \leq \left\| \nabla D([\boldsymbol{\mu}; \boldsymbol{\varepsilon}]) \right\|_2 \|[\boldsymbol{\mu}; \boldsymbol{\varepsilon}] - [\boldsymbol{\mu}'; \boldsymbol{\varepsilon}']\|_2,$$

Hence, it suffices to show $\left\| \nabla D([\boldsymbol{\mu}; \boldsymbol{\varepsilon}]) \right\|_2 \leq \eta/\sqrt{d}$ for all $[\boldsymbol{\mu}; \boldsymbol{\varepsilon}] \in V_{B_L}$. By (39), triangle inequality gives

$$\sup_{[\boldsymbol{\mu}; \boldsymbol{\varepsilon}] \in V_{B_L}} \|\nabla D([\boldsymbol{\mu}; \boldsymbol{\varepsilon}])\|_2 \leq \frac{1}{d} \sup_{[\boldsymbol{\mu}; \boldsymbol{\varepsilon}] \in V_{B_L}} \|\boldsymbol{\mu}\|_2 + \frac{1}{2d} \sup_{[\boldsymbol{\mu}; \boldsymbol{\varepsilon}] \in V_{B_L}} \|\exp(\boldsymbol{\varepsilon})\|_2 + \frac{1}{2\sqrt{d}}$$

$$\leq \frac{1}{d} B_L + \frac{1}{2\sqrt{d}} \exp(B_L) + \frac{1}{2\sqrt{d}}$$

where in the last inequality, we use $\|\boldsymbol{\mu}\|_2 \leq \|[\boldsymbol{\mu}; \boldsymbol{\varepsilon}]\|_2 \leq B_L$ and

$$\| \exp(\boldsymbol{\varepsilon}) \|_2 = \sqrt{\sum_{j=1}^{d} \exp(2\boldsymbol{\varepsilon}_j)} \leq \sqrt{d} \exp(B_L).$$

As $\frac{1}{\sqrt{d}} \leq 1$ for any $d \geq 1$, we have shown $\left\| \nabla D([\boldsymbol{\mu}; \boldsymbol{\varepsilon}]) \right\|_2 \leq \eta/\sqrt{d}$. $\qquad \square$

### B.3.4    PROOF OF THEOREM B.1

Note that $\rho_\ell$ bounded spectral norm of weight matrices and $\sigma_\ell$-Lipschitzness of activation function for each layer $\ell \in [L]$ yield

$$\sup_{f_{\boldsymbol{\Theta}} \in \mathcal{F}_{\boldsymbol{\Xi}}, \boldsymbol{x} \in \mathcal{X}} \|f_{\boldsymbol{\Theta}}(\boldsymbol{x})\|_2 \leq \underbrace{\sup_{\boldsymbol{x} \in \mathcal{X}} \|\boldsymbol{x}\|_2 \cdot \prod_{\ell=1}^{L} \rho_\ell \sigma_\ell}_{B_L}. \tag{40}$$

Therefore, for all $\boldsymbol{x} \in \mathcal{X}$, $\|[\boldsymbol{\mu}_{\boldsymbol{\Theta}}(\boldsymbol{x}); \boldsymbol{\varepsilon}_{\boldsymbol{\Theta}}(\boldsymbol{x})]\|_2 \leq B_L$ holds for all $\boldsymbol{\Theta} \in \mathcal{F}_{\boldsymbol{\Xi}}$. Then, we can apply the $\ell_2$ Lipschitzness of KL divergence in Lemma B.6 to bound the KL divergence term $D(f_{\boldsymbol{\Theta}}; \boldsymbol{x})$ in (36) as

$$
\begin{aligned}
|D(f_{\boldsymbol{\Theta}}; \boldsymbol{x}_i) - D(f_{\boldsymbol{\Theta}'}; \boldsymbol{x}_i)| &\leq \eta \cdot \|f_{\boldsymbol{\Theta}}(\boldsymbol{x}_i) - f_{\boldsymbol{\Theta}'}(\boldsymbol{x}_i)\|_2 \\
&\leq \eta \cdot \max_{i \in [N]} \|f_{\boldsymbol{\Theta}}(\boldsymbol{x}_i) - f_{\boldsymbol{\Theta}'}(\boldsymbol{x}_i)\|_2 \\
&= \frac{\eta}{\sqrt{d}} \cdot \|f_{\boldsymbol{\Theta}} - f_{\boldsymbol{\Theta}'}\|_{L_{\infty,2(\mathcal{S})}}, \qquad \forall f_{\boldsymbol{\Theta}}, f_{\boldsymbol{\Theta}'} \in \mathcal{F}_{\boldsymbol{\Xi}}.
\end{aligned}
\tag{41}
$$

Since the bound in (41) yields

$$
\begin{aligned}
\|D(f_{\boldsymbol{\Theta}}) - D(f_{\boldsymbol{\Theta}'})\|_{L_2(\mathcal{S})} &= \sqrt{\frac{1}{N} \sum_{i=1}^{N} \left( D(f_{\boldsymbol{\Theta}; \boldsymbol{x}_i}) - D(f_{\boldsymbol{\Theta}'; \boldsymbol{x}_i}) \right)^2} \\
&\leq \frac{\eta}{\sqrt{d}} \cdot \|f_{\boldsymbol{\Theta}} - f_{\boldsymbol{\Theta}'}\|_{L_{\infty,2(\mathcal{S})}}, \quad \forall f_{\boldsymbol{\Theta}}, f_{\boldsymbol{\Theta}'} \in \mathcal{F}_{\boldsymbol{\Xi}},
\end{aligned}
$$

we have

$$
\log \mathcal{N}\left(\Delta, \epsilon, L_2(\mathcal{S})\right) \leq \log \mathcal{N}\left(\mathcal{F}_{\boldsymbol{\Xi}}, \frac{\epsilon\sqrt{d}}{\eta}, L_{\infty,2}(\mathcal{S})\right),
$$

where we define $\Delta$ as the class of KL divergence terms $D(f_{\boldsymbol{\Theta}})$ in (36), i.e.,

$$
\Delta := \{D(f_{\boldsymbol{\Theta}}) : f_{\boldsymbol{\Theta}} \in \mathcal{F}_{\boldsymbol{\Xi}}\}.
$$

Now, we can apply the covering number bound in Lemma B.5 to bound the covering number $\mathcal{N}\left(\Delta, \epsilon, L_2(\mathcal{S})\right)$ as

$$
\mathcal{N}\left(\Delta, \epsilon, L_2(\mathcal{S})\right) \leq \frac{64\,\eta^2\,\overline{\mathcal{R}}_{\boldsymbol{\Xi}}^2}{\epsilon^2 d}\, \log\left(\left(\tfrac{11\eta\overline{\mathcal{R}}_{\boldsymbol{\Xi}}}{\epsilon\sqrt{d}} + 7\right) N\,W\right).
\tag{42}
$$

Define $B_\Delta := \sup_{D(f_{\boldsymbol{\Theta}}) \in \Delta} \|D(f_{\boldsymbol{\Theta}})\|_{L_2(\mathcal{S})}$. We further bound the empirical Rademacher complexity $\widehat{R}_{\mathcal{S}}(\Delta)$ by applying the Dudley's entropy integral Lemma B.4 with the bound on covering number (42) by the choosing $\alpha = \frac{1}{\sqrt{N}}$:

$$
\begin{aligned}
\widehat{\mathfrak{R}}_{\mathcal{S}}(\Delta) &\leq \frac{4}{\sqrt{N}} + \frac{12}{\sqrt{N}} \int_{\frac{1}{\sqrt{N}}}^{B_\Delta} \sqrt{\log\left(\mathcal{N}(\mathcal{D}, \epsilon, L_2(\mathcal{S}))\right)}\, d\epsilon \\
&\leq \frac{4}{\sqrt{N}} + \frac{96\eta\overline{\mathcal{R}}_{\boldsymbol{\Xi}}}{\sqrt{Nd}} \sqrt{\log\left(\left(\frac{11\eta\overline{\mathcal{R}}_{\boldsymbol{\Xi}}\sqrt{N}}{\sqrt{d}} + 7\right) NW\right)} \int_{\frac{1}{\sqrt{N}}}^{B_\Delta} \frac{1}{\epsilon}\, d\epsilon \\
&\overset{(42)}{\leq} \frac{4}{\sqrt{N}} + \frac{96\eta\overline{\mathcal{R}}_{\boldsymbol{\Xi}}}{\sqrt{Nd}} \sqrt{\log\left(\left(\frac{11\eta\overline{\mathcal{R}}_{\boldsymbol{\Xi}}\sqrt{N}}{\sqrt{d}} + 7\right) NW\right)} \log(\sqrt{N}B_\Delta).
\end{aligned}
\tag{43}
$$

Now, we apply the Rademacher generalization bound Lemma B.3 with the bound on the empirical Rademacher complexity in (43) by letting $M = \sup_{\boldsymbol{\Theta} \in \boldsymbol{\Xi}, \boldsymbol{x} \in \mathcal{X}} |D(f_{\boldsymbol{\Theta}}; \boldsymbol{x})|$. Hence, it holds that

$$
\begin{aligned}
\sup_{f_{\boldsymbol{\Theta}} \in \mathcal{F}_{\boldsymbol{\Xi}}} &\left| \mathbb{E}_{\boldsymbol{x} \sim \mathcal{D}}\left[D(f_{\boldsymbol{\Theta}}; \boldsymbol{x})\right] - \frac{1}{N} \sum_{i=1}^{N} D(f_{\boldsymbol{\Theta}}(\boldsymbol{x}_i)) \right| \\
&\leq 2\widehat{\mathfrak{R}}_{\mathcal{S}}(\Delta) + 3M\sqrt{\frac{\log 2/\delta}{2N}} \\
&\leq \frac{8}{\sqrt{N}} + \frac{192\eta\overline{\mathcal{R}}_{\boldsymbol{\Xi}}}{\sqrt{Nd}} \sqrt{\log\left(\left(\frac{11\eta\overline{\mathcal{R}}_{\boldsymbol{\Xi}}\sqrt{N}}{\sqrt{d}} + 7\right) NW\right)} \log(\sqrt{N}B_\Delta) + 3M\sqrt{\frac{\log 2/\delta}{2N}},
\end{aligned}
\tag{44}
$$

with probability at least $1 - \delta$.

Suppose that the remainder of the proof is conditioned on the event (44). We will now show the upper bounds on $B_\Delta$, $M$, and $\overline{\mathcal{R}}_{\boldsymbol{\Xi}}$ in (44).

**Bound on $B_\Delta$:**   Using $\ell_2$ lipschitzness of $D(f_\Theta; \boldsymbol{x}_i)$ in Lemma B.6 gives

$$|D(f_\Theta; \boldsymbol{x}_i)| \leq \eta \|f_\Theta(\boldsymbol{x}_i)\|_2, \quad \forall f_\Theta \in \mathcal{F}_\Xi, \quad \forall \boldsymbol{x}_i \in \mathcal{S}.$$

This directly gives us

$$B_\Delta = \sup_{D(f_\Theta) \in \Delta} \sqrt{\frac{1}{N} \sum_{i=1}^N \left(D(f_\Theta; \boldsymbol{x}_i)\right)^2} \leq \sqrt{\frac{\eta^2}{N} \sum_{i=1}^N \sup_{\Theta \in \Xi} \|f_\Theta(\boldsymbol{x}_i)\|_2^2} \tag{45}$$
$$\leq \eta B_L,$$

where $B_L$ is defined in (40).

**Bound on $M$:**   Upper bound on $M$ can be obtained in a similar way as the bound on $B_\Delta$ in (45):

$$M = \sup_{\Theta \in \Xi, \boldsymbol{x} \in \mathcal{X}} |D(f_\Theta; \boldsymbol{x})| \leq \eta \sup_{\Theta \in \Xi, \boldsymbol{x} \in \mathcal{X}} \|f_\Theta(\boldsymbol{x})\|_2 \leq \eta B_L. \tag{46}$$

**Bound on $\overline{\mathcal{R}}_\Xi$:**   In the proof of (Hieu et al., 2025, Theorem 1), Hieu et al. show an upper bound on $\overline{\mathcal{R}}_\Xi$ which is

$$\overline{\mathcal{R}}_\Xi \leq \sup_{\boldsymbol{x} \in \mathcal{X}} \|\boldsymbol{x}\|_2 \prod_{\ell=1}^L \sigma_\ell \rho_\ell \left(\sum_{\ell'=1}^L (a_{\ell'}/\rho_{\ell'})^{2/3}\right)^{3/2}. \tag{47}$$

For the sake of simplicity, we denote $B_x = \sup_{\boldsymbol{x} \in \mathcal{X}} \|\boldsymbol{x}\|_2$. Putting the bounds (45), (46), and (47) into (44) provides

$$\sup_{f_\Theta \in \mathcal{F}_\Xi} \left| \mathbb{E}_{\boldsymbol{x} \sim \mathcal{D}} \left[D(f_\Theta; \boldsymbol{x})\right] - \frac{1}{N} \sum_{i=1}^N D(f_\Theta(\boldsymbol{x}_i)) \right|$$

$$\leq \tilde{\mathcal{O}}\left( \frac{\eta}{\sqrt{N}} \sqrt{\log(W)} B_x \prod_{\ell=1}^L \sigma_\ell \rho_\ell \left(\sum_{\ell'=1}^L (a_{\ell'}/\rho_{\ell'})^{3/2}\right)^{3/2} + \eta B_x \prod_{\ell=1}^L (\rho_\ell \sigma_\ell) \sqrt{\frac{\log(1/\delta)}{N}} \right)$$

$$\leq \tilde{\mathcal{O}}\left( \frac{\eta B_x \prod_{\ell=1}^L \sigma_\ell \rho_\ell}{\sqrt{N}} \left[ \sqrt{\log(W)} \left(\sum_{\ell'=1}^L (a_{\ell'}/\rho_{\ell'})^{3/2}\right)^{3/2} \vee \sqrt{\log(1/\delta)} \right] \right).$$

# C   DISCUSSION ON THE APPROXIMATION IN SECTION 3.1

## C.1   DISCUSSION ON (10)

The key step in our decoder-free ELBO maximization is the approximation

$$\mathbb{E}_{q_\theta(\boldsymbol{z}|\boldsymbol{x})}\left[\log p(\boldsymbol{x}|\boldsymbol{z})\right] \approx \mathbb{E}_{q_\theta(\boldsymbol{z}|\boldsymbol{x}) q_\theta(\boldsymbol{z}'|\boldsymbol{x})}\left[\log p(\boldsymbol{z}'|\boldsymbol{z})\right] \tag{48}$$

**Lower-bound view.**   As shown in Lemma 3.1, this approximation admits a lower bound up to an additive constant independent of $\boldsymbol{z}$:

$$\mathbb{E}_{q_\theta(\boldsymbol{z}|\boldsymbol{x})}[\log p(\boldsymbol{x}|\boldsymbol{z})] \geq \mathbb{E}_{q_\theta(\boldsymbol{z}|\boldsymbol{x}) q_\theta(\boldsymbol{z}'|\boldsymbol{x})}[\log p(\boldsymbol{z}'|\boldsymbol{z})] + \text{const.} \tag{49}$$

Consequently, maximizing the right-hand side with respect to $\theta$ implicitly maximizes the reconstruction term $\mathbb{E}_{q_\theta(\mathbf{z}|\mathbf{x})}\left[\log p(\mathbf{x} \mid \mathbf{z})\right]$, which is the objective of ELBO maximization. Moreover, using (12) (see Section 3.1), the surrogate is negatively related to InfoNCE:

$$\mathbb{E}_{q_\theta(\mathbf{z}|\mathbf{x})}\left[\log p(\mathbf{x} \mid \mathbf{z})\right] \approx -I_{\text{NCE}}(\mathbf{x}; \mathbf{x}'), \tag{50}$$

so minimizing the InfoNCE loss increases the reconstruction term.

**Change-of-variables view.** Another perspective on the reconstruction approximation (10) comes from a change of variables. Let $g$ be an invertible, differentiable mapping such that $\boldsymbol{x} = g(\boldsymbol{z}')$. Then, by the change-of-variables formula,

$$p(\boldsymbol{x} \mid \boldsymbol{z}) = p(\boldsymbol{z}' \mid \boldsymbol{z}) \left|\det J_{g^{-1}}(\boldsymbol{x})\right| = p(\boldsymbol{z}' \mid \boldsymbol{z}) \left|\det J_g(\boldsymbol{z}')\right|^{-1}, \tag{51}$$

where $J_g$ and $J_{g^{-1}}$ denote the Jacobians of $g$ and $g^{-1}$, respectively, and $\boldsymbol{z}' = g^{-1}(\boldsymbol{x})$. Taking logarithms yields

$$\log p(\boldsymbol{x} \mid \boldsymbol{z}) = \log p(\boldsymbol{z}' \mid \boldsymbol{z}) + \log\left|\det J_{g^{-1}}(\boldsymbol{x})\right| = \log p(\boldsymbol{z}' \mid \boldsymbol{z}) - \log\left|\det J_g(\boldsymbol{z}')\right|, \tag{52}$$

where the second term depends only on $\boldsymbol{x}$ (equivalently, on $\boldsymbol{z}'$) and is independent of $\boldsymbol{z}$.

**Sufficient condition (tightness).** If, in addition to invertibility, $g$ is *volume-preserving*, i.e., $\left|\det J_{g^{-1}}(\boldsymbol{x})\right| \equiv 1$ (equivalently, $\left|\det J_g(\boldsymbol{z}')\right| \equiv 1$) on the data manifold, then the additive term in (52) vanishes and we obtain the tight equality $\log p(\boldsymbol{x} \mid \boldsymbol{z}) = \log p(\boldsymbol{z}' \mid \boldsymbol{z})$. More generally, when $\left|\det J_{g^{-1}}(\boldsymbol{x})\right|$ is approximately constant over the data manifold, the additive term acts as (approximately) a constant shift independent of $\boldsymbol{z}$, yielding a tight surrogate for optimization.

This assumption is plausible in practice under the commonly observed *dimension-collapse* phenomenon: the embeddings $\boldsymbol{z}'$ have effective rank (intrinsic dimension) much smaller than the ambient embedding dimension yet retain nearly all task-relevant information about the features $\boldsymbol{x}$. When the feature and embedding manifolds have (approximately) the same intrinsic dimension and $g$ behaves near-isometrically between them, the Jacobian determinant varies weakly, making the surrogate in (52) tight in practice.

### C.2 Gaussian KL Surrogate for Projected-Normal KL

We study the tightness of the bound in (16), repeated here:

$$D\big(\mathcal{N}(\mu, K) \,\|\, \mathcal{N}(0, I_d)\big) \ \geq \ D\big(\mathcal{PN}(\mu, K) \,\|\, \mathrm{Unif}(\mathcal{S}^{d-1})\big). \tag{53}$$

Before analyzing tightness, we note several practical benefits of using the Gaussian KL as a surrogate for the projected-normal KL:

- **Closed form.** It is trivial to implement and numerically stable.
- **Aligned optima.** The Gaussian KL and projected-normal KL share the same minimizer (e.g., at $\mu = 0$ and $K = I_d$), so optimizing the surrogate steers the model toward the same optimum.
- **Efficiency.** Unlike Monte Carlo or $k$-NN estimators needed for the projected-normal KL, the Gaussian KL requires no sampling.

Moreover, the KL term acts only as a regularizer, whereas InfoNCE directly drives semantic similarity; thus modest approximation error in the KL has limited effect on downstream performance.

We assess tightness by comparing the closed-form Gaussian KL with an estimated projected-normal KL using a divergence estimator (Wang et al., 2009) in two settings: synthetic data and CIFAR-10 under VCL training.

**KL gap on synthetic data.** We approximate $D\big(\mathcal{PN}(\mu, K) \,\|\, \mathrm{Unif}(\mathcal{S}^{d-1})\big)$ numerically using $10^5$ samples in dimension $d = 128$ for random $(\mu, K)$ draws, with $\mu \sim \mathcal{N}(0, I_d)$ and

$$K = \tfrac{1}{d} A A^\top + 0.1\, I_d, \quad A_{ij} \sim \mathcal{N}(0, 0.5) \ \forall i, j. \tag{54}$$

We employ the $k$-nearest-neighbor divergence estimator (Wang et al., 2009) with $k = 1$, compute both the Gaussian KL (analytically) and the projected-normal KL (using the estimator) on the same samples, and repeat over 20 random trials to reduce variance.

Table 2 reports the gap between the two KLs on synthetic data: the average absolute gap is approximately $9.49$ (about a $10\%$ relative difference). Thus, the Gaussian KL surrogate closely tracks the projected-normal KL while retaining the practical advantages noted above.

Table 2: Gaussian KL (G-KL) vs. projected normal KL (PN-KL) on synthetic data.

|      | G-KL   | PN-KL | Gap (G-KL−PN-KL) | Ratio (G-KL/PN-KL) |
|------|--------|-------|------------------|--------------------|
| mean | 106.86 | 97.37 | 9.49             | 0.91               |
| std  | 9.56   | 7.63  | -                | -                  |

**KL gap on CIFAR-10.** Beyond the synthetic study, we measure the gap during VCL training on CIFAR-10 using the same experimental settings (Appendix D.1); results are shown in Figure 6. After only a few epochs, the Gaussian KL and the projected-normal KL closely track each other. This indicates that minimizing the Gaussian-KL surrogate effectively minimizes the projected-normal KL—the quantity we aim to reduce—while retaining the practical advantages of the surrogate.

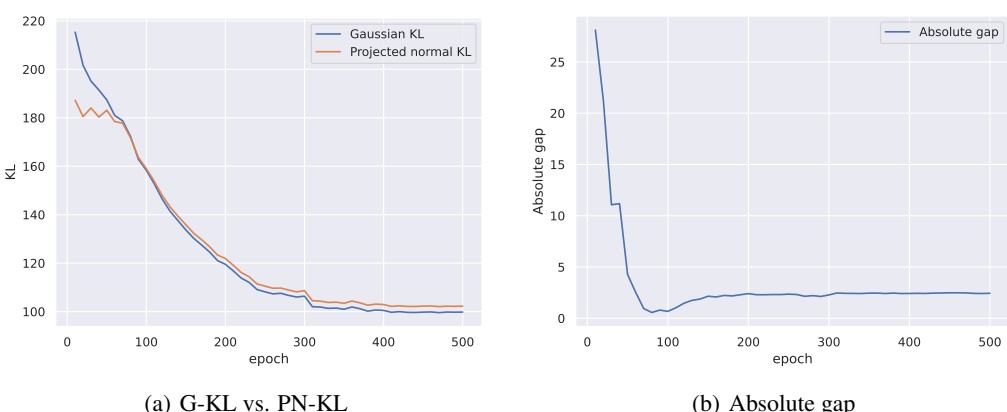

(a) G-KL vs. PN-KL

(b) Absolute gap

Figure 6: Tracking Gaussian KL (G-KL) and projected normal KL (PN-KL) during VCL training on CIFAR-10. (a) G-KL vs. PN-KL; (b) Absolute gap, |G-KL − PN-KL|. This shows that minimizing Gaussian KL leads to minimizing projected normal KL.

## D EXPERIMENTS

### D.1 TRAINING DETAILS AND HYPERPARAMETERS

**Datasets and preprocessing.** Experiments are conducted on CIFAR-10 (Krizhevsky et al., 2009), CIFAR-10C (Hendrycks & Dietterich, 2019), CIFAR-10H (Peterson et al., 2019), CIFAR-100 (Krizhevsky et al., 2009), STL-10 (Coates et al., 2011), Tiny-ImageNet (Le & Yang, 2015), and Caltech-256 (Griffin et al., 2007). Following SimCLR, we sample two views per image via random resized crop (image size $32 \times 32$ and scale $[0.2, 1.0]$), horizontal flip ($p$=0.5), color jitter (brightness/contrast/saturation/hue = 0.4, applied with $p$=0.8), Gaussian blur (kernel size 9), and random grayscale ($p$=0.2). Inputs are normalized with dataset-specific means/standard deviations.

**Architectures.** Encoders are ResNet-18, with embedding dimension $d$=128.

**Optimization.** We use AdamW (Loshchilov & Hutter, 2019) with base LR $10^{-2}$ (encoder and head), weight decay $10^{-4}$, batch size $B$=512, and $T$=500 epochs for pretraining and $T = 100$ for training linear classifier. Temperature for InfoNCE loss is $\tau$=0.07. We set $m$=1 posterior samples per view for VSimCLR and VSupCon by default (ablation in Table 4). No momentum encoder or queue is used; all negatives are in-batch. For training stability, we clip the posterior log-variance ($\log \boldsymbol{\sigma^2}$) to $[-5, 5]$ to bound variances, and clip gradient global norm at 1.0.

Table 3: Log-determinant of average posterior covariance $K$ for each CIFAR-10 class.

| Index | Class | $\log\det(K)$ |
|---|---|---|
| 0 | airplane | -182.207 |
| 1 | automobile | -181.691 |
| 2 | bird | -183.713 |
| 3 | cat | -191.317 |
| 4 | deer | -184.969 |
| 5 | dog | -185.432 |
| 6 | frog | -182.125 |
| 7 | horse | -179.331 |
| 8 | ship | -185.991 |
| 9 | truck | -188.179 |

## D.2 ADDITIONAL RESULTS ON DIMENSION COLLAPSE

In addition to the singular spectrum of VCL embeddings on CIFAR-10 and CIFAR-100 in Figure 3, Figure 7 reports results on Caltech-256 and Tiny-ImageNet. In both datasets, VCL mitigates the dimension-collapse phenomenon commonly observed in contrastive learning.

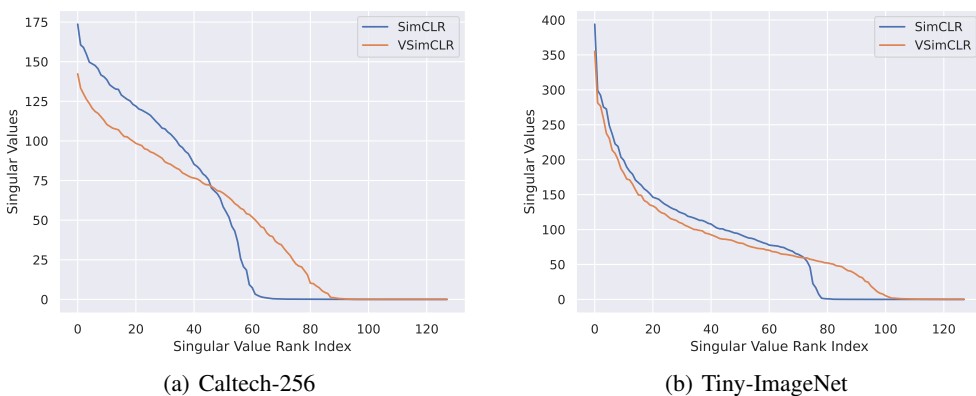

(a) Caltech-256        (b) Tiny-ImageNet

Figure 7: Singular-value spectrum of the embedding covariance on Cartech-256 and Tiny-ImageNet. VSimCLR mitigates dimensional collapse on both datasets.

## D.3 DISTRIBUTIONAL CONTRASTIVE LOSS

In addition to the contrastive loss on embeddings, it is worthwhile to contrast the posterior distributions within the VCL framework. Specifically, we aim to pull together the posteriors corresponding to different augmentations of the same input and to push apart posteriors from distinct inputs. To incorporate this into VCL, we introduce the *DistNCE* loss, a contrastive objective over posterior parameters, defined as

$$D_{\text{DistNCE}}(\theta) = -\mathbb{E}\left[\log \frac{\exp\big(s(\theta, \theta^+)\big)}{\sum_j \exp\big(s(\theta, \theta_j)\big)}\right], \tag{55}$$

where $\theta$ denotes the posterior parameters $(\boldsymbol{\mu}, K)$, $\theta^+$ is the positive-pair parameter for the same input, and $\{\theta_j\}_{j\neq+}$ are negative-pair parameters from other inputs. The expectation is taken over the joint distribution $p(\theta, \theta^+)\prod_{j\neq+} p(\theta_j)$.

Moreover, we increase the number of posterior samples used for the InfoNCE loss. Specifically, we draw $m$ samples $\{\mathbf{z}^{(k)}\}_{k=1}^m$ from each posterior, resulting in an $m$-fold increase in effective batch

Table 4: Classification accuracy on STL10 with different number of embedding generation from posterior. We report top-1 and top-5 accuracies of SimCLR, VSimCLR, SupCon, and VSupCon across the datasets with different $m$ and DistNCE (55).

| METHOD | STL10 | |
|---|---|---|
| | TOP1 | TOP5 |
| SIMCLR | 60.44 | 95.80 |
| VSIMCLR ($m = 1$) | 60.11 | 92.00 |
| VSIMCLR ($m = 4$) | 57.86 | 88.29 |
| VSIMCLR ($m = 16$) | 59.13 | 92.85 |
| VSIMCLR ($m = 64$) | 56.91 | 86.63 |
| VSIMCLR WITH DISTNCE (55) | 36.54 | 80.25 |
| VSIMCLR (ASYM) | 57.38 | 88.78 |
| SUPCON | 75.88 | 75.88 |
| VSUPCON ($m = 1$) | 75.76 | 96.99 |
| VSUPCON ($m = 4$) | 74.35 | 97.14 |
| VSUPCON ($m = 16$) | 76.11 | 98.39 |
| VSUPCON ($m = 64$) | 77.96 | 98.44 |

size, and compute the InfoNCE loss over this enlarged set of embeddings. The classification results are reported in Table 4.

We also evaluate the performance of the asymmetric lower bound (18) (denoted ASYM) in place of the symmetrized objective (19). These results are also shown in Table 4.

From these experiments, we did not observe any significant differences when applying DistNCE (55), using the asymmetric loss, or sampling multiple embeddings per posterior. Based on these findings, we proceed with the basic VCL variants from the main text for all subsequent experiments.

### D.4 EFFECT OF KL REGULARIZER ON CLASSIFICATION

As shown in Table 1, VSupCon exhibits reduced classification accuracy on some datasets, whereas VSimCLR remains stable. We attribute this degradation to two factors:

1. VSimCLR's objective coincides with the VCL objective in (19), but VSupCon's does not, creating a mismatch that can impede proper ELBO maximization.

2. SupCon optimizes embeddings directly for classification; adding a KL term can conflict with this objective.

We therefore hypothesize that weakening the KL regularizer improves VSupCon's accuracy. To test this, we scale the KL term by $\beta \in \{1, 10^{-1}, 10^{-2}, 10^{-3}\}$,

$$\mathcal{L}^{\text{vsup}}(\beta) = \mathcal{L}^{\text{sup}} + \beta D_{\text{KL}}\big(q_\theta(\boldsymbol{z} \mid \boldsymbol{x}) \,\|\, p(\boldsymbol{z})\big), \tag{56}$$

and evaluate the resulting embeddings. In Table 5, as expected, smaller $\beta$ (i.e., a weaker KL effect) yields higher accuracy. Thus, for pure classification tasks, SupCon may not benefit from a VCL variant unless the KL weight is carefully tuned.

### D.5 IMPLICATIONS OF DISTRIBUTIONAL EMBEDDINGS

Distributional (probabilistic) embeddings provide useful capabilities, including uncertainty quantification and probability-based distances between samples and classes. We analyze them along three axes: uncertainty, typicality, and out-of-distribution (OOD) behavior.

**Posterior covariance vs. uncertainty.** As shown in Figure 8, different samples exhibit varying degrees of posterior dispersion (e.g., the log-determinant of the covariance, $\log \det(K)$), which can serve as an uncertainty measure. To examine how uncertainty and posterior covariance are related, we conduct experiments on two benchmark datasets, CIFAR-10H (Peterson et al., 2019) and CIFAR-10C (Hendrycks & Dietterich, 2019):

Table 5: Classification accuracy on STL10 with different number of embedding generation from posterior. We report top-1 and top-5 accuracies of SimCLR, VSimCLR, SupCon, and VSupCon across the datasets with different $m$ and DistNCE (55).

| $\beta$ | TOP-1 ACCURACY | TOP-5 ACCURACY |
|---|---|---|
| 1 | 47.90 | 72.34 |
| 0.1 | 47.24 | 71.90 |
| 0.01 | 50.35 | 73.27 |
| 0.001 | 51.34 | 73.09 |

Figure 8: Sample images from the CIFAR-10, organized by class (columns) and sorted by their corresponding $\log \det(K)$ (rows). In each column, the top image has the highest $\log \det(K)$, the bottom image the lowest; the overlaid numbers indicate each image's $\log \det(K)$.

- **CIFAR-10H:** The test set provides soft labels (Ishida et al., 2023; Jeong et al., 2023; 2024a) aggregated from multiple annotators. Using these soft labels, we compute the per-sample label entropy as a measure of uncertainty about the underlying class.
- **CIFAR-10C:** The test set provides systematically corrupted images with multiple corruption types and severities (higher severity = stronger corruption), which induces greater label ambiguity and thus higher uncertainty.

Beyond comparing $\log \det(K)$ with label entropy in Figure 9, we also compare the trace of $K$ (denoted $\operatorname{tr}(K)$) against label entropy in Figure 10. In both cases, we observe a *negative* slope under a first-order linear fit. This indicates that VSimCLR assigns **lower** posterior dispersion to inputs with greater label uncertainty. Conversely, inputs that humans classify unambiguously—i.e., prototypical class examples—exhibit posteriors with **larger** dispersion, suggesting their latent representations span a broader region of the class-specific embedding space; ambiguous or outlier inputs yield **smaller** dispersion, reflecting more concentrated latent distributions.

A similar pattern appears in Figures 5 and 11, which relate $\log \det(K)$ to corruption severity on CIFAR-10C. We train VSimCLR and VSupCon on CIFAR-10 and evaluate their embeddings on CIFAR-10C. Because higher severity entails stronger corruption and greater label ambiguity, these figures further support the finding that posterior covariance dispersion is negatively correlated with uncertainty. Tables 7 and 8 report the mean $\log \det(K)$ for each corruption type and severity level.

This counterintuitive observation—that typical (i.e., common) samples exhibit larger posterior dispersion—parallels the concurrent findings of Guth et al. (Guth et al., 2025), albeit under different settings: (i) **Quantity:** we analyze latent-space posterior dispersion via $\log \det K$, whereas they study input-space marginal density $p(x)$; (ii) **Observation:** typical samples have larger $\log \det K$ (ours), while they have lower $p(x)$ (theirs). Although these quantities live in different spaces, both results

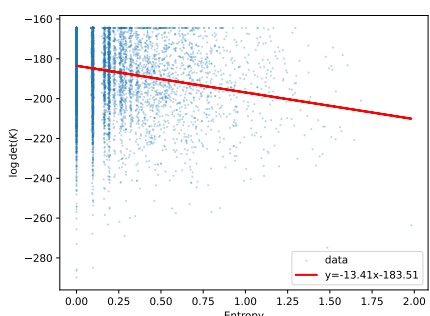

Figure 9: Posterior dispersion versus label ambiguity. Each point plots $\log \det(K)$ against the entropy of class probabilities from CIFAR-10H, with a first-order linear fit (red line).

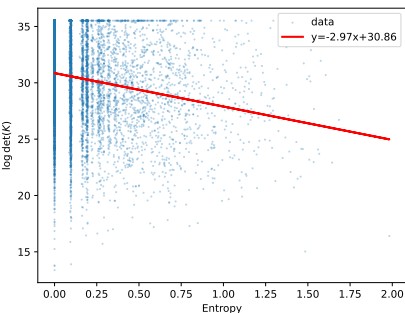

Figure 10: Relationship between posterior dispersion and label ambiguity. Each point plots the trace of $K$ ($\text{tr}(K)$) against the entropy of human-annotated class probabilities from CIFAR-10H (Peterson et al., 2019), with a first-order linear fit (red line). Similar to the result in Figure 9, the dispersion is negatively correlated with label ambiguity.

indicate that typical samples are not the highest-density points. In our case, typical images yield larger dispersion and atypical images smaller dispersion; since dispersion is inversely related to peak density, our result is consistent with Guth et al. Hence, in both settings, "typical" $\neq$ "highest-density." Consequently, posterior dispersion serves as a useful uncertainty signal; see Table 6 for an application under label scarcity.

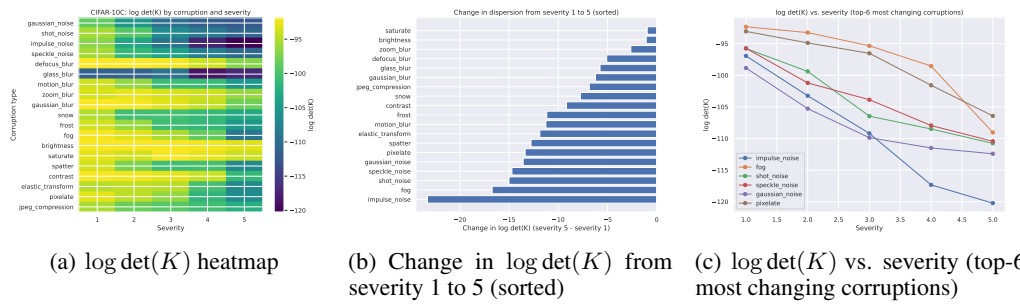

(a) $\log \det(K)$ heatmap

(b) Change in $\log \det(K)$ from severity 1 to 5 (sorted)

(c) $\log \det(K)$ vs. severity (top-6 most changing corruptions)

Figure 11: $\log \det(K)$ of **VSupCon** embeddings on CIFAR-10C (Hendrycks & Dietterich, 2019) under different corruption types and severities. "Severity" denotes the corruption level. The observed negative correlation between $\log \det(K)$ and severity is consistent with our finding that more uncertain samples exhibit smaller posterior covariance dispersion. Exact $\log \det(K)$ values are in Table 8.

Table 6: Classification accuracy on CIFAR-100 with label scarcity. We use ResNet-18 back-bone and same augmentations for all experiments. We sample the labelled subset once and report the mean accuracy of five runs with (standard error).

| METHODS | 1 LABELS / CLASS | 3 LABELS / CLASS | 5 LABELS / CLASS | 10 LABELS / CLASS | 20 LABELS / CLASS |
|---|---|---|---|---|---|
| SIMCLR | 12.22 (0.12) | 21.37 (0.15) | 26.37 (0.01) | 33.09 (0.11) | 38.00 (0.06) |
| VSIMCLR | 15.57 (0.09) | 25.70 (0.19) | 30.89 (0.11) | 37.40 (0.08) | 42.13 (0.10) |
| VSIMCLR+WT | **15.97** (0.08) | **26.07** (0.20) | **31.12** (0.06) | **37.48** (0.08) | **42.36** (0.03) |
| SUPCON | 71.55 (0.04) | 71.56 (0.05) | 71.64 (0.02) | 71.65 (0.03) | 72.07 (0.05) |
| VSUPCON | 71.77 (0.12) | **71.79** (0.10) | **71.96** (0.09) | **72.07** (0.05) | **72.16** (0.04) |
| VSUPCON+WT | **71.87** (0.02) | 71.78 (0.07) | **71.94** (0.07) | **72.07** (0.07) | **72.16** (0.06) |

Table 7: Average $\log \det K$ of VSimCLR embeddings on CIFAR-10C for each corruption type and severity (higher severity = stronger corruption).

| Corruption | Severity 1 | Severity 2 | Severity 3 | Severity 4 | Severity 5 |
|---|---|---|---|---|---|
| gaussian_noise | -187.74 | -189.85 | -192.23 | -193.05 | -193.70 |
| shot_noise | -187.49 | -188.11 | -190.18 | -190.95 | -191.97 |
| impulse_noise | -188.25 | -190.71 | -192.61 | -194.66 | -194.82 |
| speckle_noise | -187.59 | -188.64 | -189.21 | -189.93 | -190.48 |
| defocus_blur | -184.41 | -183.84 | -182.67 | -187.67 | -186.76 |
| glass_blur | -192.35 | -191.76 | -192.03 | -194.36 | -193.98 |
| motion_blur | -185.83 | -187.53 | -189.88 | -189.78 | -191.94 |
| zoom_blur | -185.95 | -183.85 | -183.86 | -183.75 | -185.07 |
| gaussian_blur | -184.43 | -182.83 | -182.11 | -183.47 | -191.56 |
| snow | -186.92 | -189.86 | -190.48 | -193.08 | -193.89 |
| frost | -188.43 | -190.13 | -192.08 | -192.16 | -193.85 |
| fog | -185.61 | -187.61 | -189.65 | -193.37 | -204.82 |
| brightness | -184.89 | -185.43 | -186.17 | -187.16 | -189.70 |
| saturate | -186.40 | -191.14 | -185.02 | -186.36 | -187.87 |
| spatter | -186.32 | -188.43 | -191.12 | -188.88 | -191.03 |
| contrast | -185.67 | -188.03 | -189.84 | -192.59 | -200.25 |
| elastic_transform | -185.66 | -185.12 | -184.95 | -189.66 | -195.31 |
| pixelate | -185.10 | -186.44 | -187.62 | -188.58 | -189.46 |
| jpeg_compression | -182.94 | -183.30 | -183.73 | -184.38 | -185.28 |

**Class-wise average posterior parameters.** Figure 12 reports class-wise averages of the posterior parameters—the mean norm $\|\boldsymbol{\mu}\|$ and the covariance dispersion $\log \det K$—for VSimCLR and VSupCon. Classes exhibit distinct dispersion profiles. Despite being trained independently, the two methods yield similar class-wise patterns in both quantities: for example, the *cat* and *dog* classes show comparatively lower $\|\boldsymbol{\mu}\|$ and $\log \det K$, whereas *truck* attains the largest $\|\boldsymbol{\mu}\|$. Table 3 provides detailed per-class $\log \det K$ values.

**Posterior on in-distribution vs. out-of-distribution.** We compare per-sample posterior parameters under VSimCLR for in-distribution (ID; CIFAR-10) versus out-of-distribution (OOD; SVHN (Netzer et al., 2011)) inputs. VSimCLR is trained on the CIFAR-10 training set, after which we extract $(\boldsymbol{\mu}, K)$ on the CIFAR-10 and SVHN test sets. Figure 13 plots the pairs $(\|\boldsymbol{\mu}\|, \log \det K)$ for each dataset; black markers denote dataset-wise means. While the mean values $\text{avg}(\|\boldsymbol{\mu}\|)$ and $\text{avg}(\log \det K)$ are similar across CIFAR-10 and SVHN, the SVHN points exhibit substantially greater spread (dispersion) across samples, indicating a broader posterior-parameter distribution for OOD data.

Table 8: Average $\log\det K$ of VSupCon embeddings on CIFAR-10C for each corruption type and severity (higher severity = stronger corruption).

| Corruption | Severity 1 | Severity 2 | Severity 3 | Severity 4 | Severity 5 |
|---|---|---|---|---|---|
| gaussian_noise | -98.85 | -105.28 | -109.87 | -111.50 | -112.42 |
| shot_noise | -95.76 | -99.39 | -106.47 | -108.50 | -110.77 |
| impulse_noise | -96.94 | -103.24 | -109.20 | -117.34 | -120.23 |
| speckle_noise | -95.73 | -101.21 | -103.87 | -107.95 | -110.44 |
| defocus_blur | -91.95 | -91.90 | -92.33 | -93.94 | -97.03 |
| glass_blur | -111.32 | -111.29 | -109.63 | -118.74 | -117.08 |
| motion_blur | -93.95 | -96.48 | -100.86 | -100.96 | -105.21 |
| zoom_blur | -93.66 | -92.94 | -93.67 | -94.06 | -96.29 |
| gaussian_blur | -91.95 | -92.31 | -93.14 | -94.40 | -98.17 |
| snow | -95.28 | -100.62 | -100.32 | -101.30 | -103.04 |
| frost | -93.98 | -96.23 | -100.71 | -101.33 | -105.15 |
| fog | -92.33 | -93.25 | -95.34 | -98.54 | -109.05 |
| brightness | -92.04 | -92.06 | -92.16 | -92.40 | -93.11 |
| saturate | -93.05 | -93.80 | -92.14 | -92.82 | -94.02 |
| spatter | -93.86 | -97.46 | -100.59 | -100.27 | -106.63 |
| contrast | -92.14 | -92.54 | -93.10 | -94.30 | -101.31 |
| elastic_transform | -95.01 | -94.65 | -94.96 | -100.26 | -106.89 |
| pixelate | -93.06 | -94.88 | -96.53 | -101.58 | -106.43 |
| jpeg_compression | -95.47 | -98.31 | -99.28 | -100.59 | -102.32 |

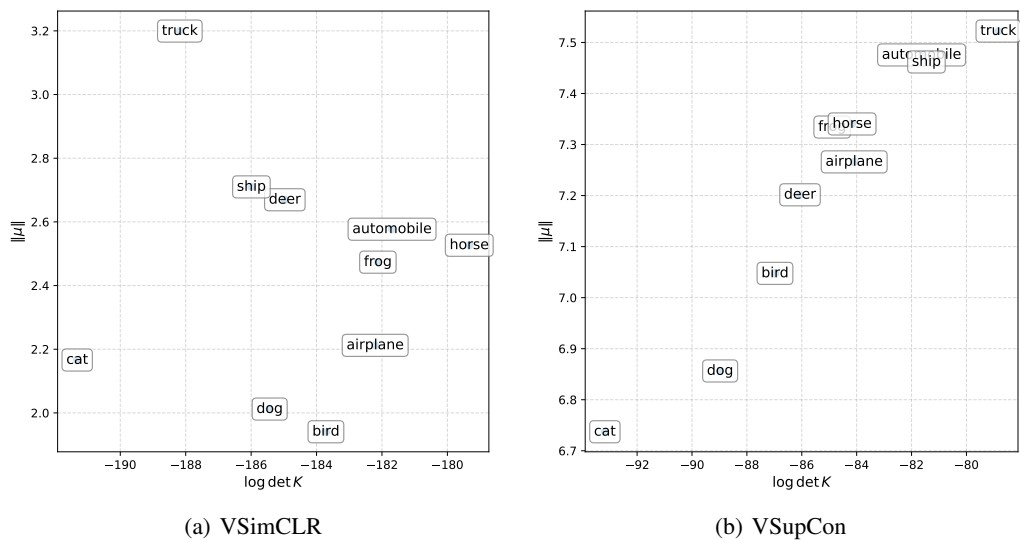

(a) VSimCLR

(b) VSupCon

Figure 12: Norm of the posterior mean $\|\boldsymbol{\mu}\|$ versus the log-determinant of the covariance $\log\det(K)$, averaged per class. Both $\boldsymbol{\mu}$ and $K$ are computed by averaging over all samples belonging to the same class.

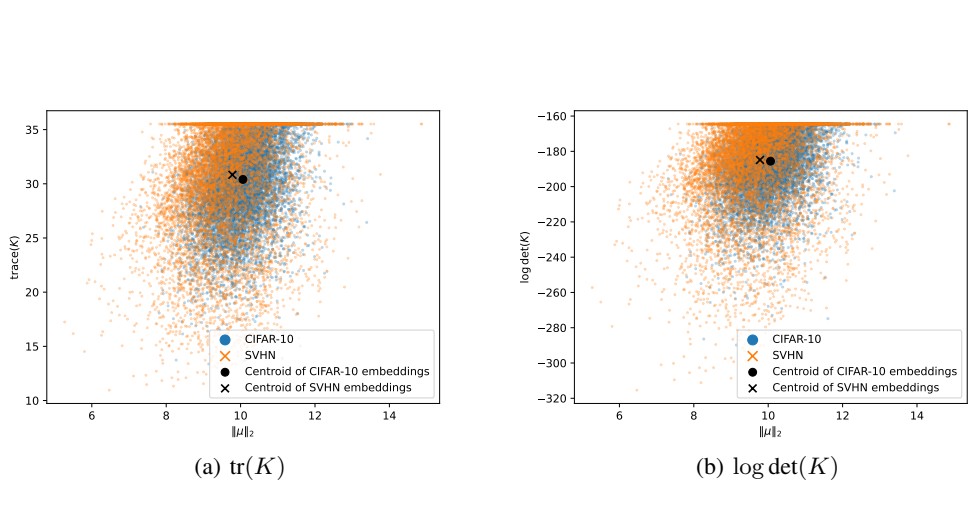

(a) $\mathrm{tr}(K)$

(b) $\log \det(K)$

Figure 13: Posterior parameters of CIFAR-10 and SVHN datasets. We use the same encoder of VSimCLR trained with CIFAR-10.

