# OpenReview forum: "Probabilistic Variational Contrastive Learning"
_ICLR.cc/2026/Conference — ICLR 2026 Conference Withdrawn Submission_

### Official Review · Reviewer_AjyY · 2025-10-29

**Soundness:** 1
**Presentation:** 3
**Contribution:** 2
**Rating:** 2
**Confidence:** 4

**Summary:**

The paper introduces **Variational Contrastive Learning (VCL)**, a probabilistic framework that reformulates contrastive learning as decoder-free ELBO maximization. The key idea is to interpret the InfoNCE loss as a surrogate reconstruction term and introduce a KL divergence regularizer between the approximate posterior $q_\theta(z|x)$ (modeled as a projected normal distribution) and a uniform prior on the unit hypersphere. This enables learning **probabilistic embeddings** that quantify uncertainty. Two instantiations, **VSimCLR** and **VSupCon**, extend SimCLR and SupCon respectively by replacing deterministic embeddings with samples from the learned posterior. The paper includes theoretical results connecting InfoNCE and ELBO, a generalization bound for the KL term, and experiments showing reduced dimensional collapse and improved uncertainty calibration, while matching or slightly improving classification accuracy.

**Strengths:**

1. The topic is highly relevant and timely, as **uncertainty modeling** and **probabilistic embeddings** are increasingly important in self-supervised and contrastive learning research.
2. The proposed approach is conceptually simple, easy to implement, and broadly compatible with existing contrastive frameworks without requiring architectural changes.
3. The paper is **clearly written** and well organized, with helpful figures, examples, and ablation studies that aid understanding.
4. The analysis in **Section 4.5**, which relates the magnitude of the posterior covariance matrix to predictive uncertainty, is interesting and relatively unexplored in prior literature, representing a potentially valuable empirical insight.

**Weaknesses:**

## Major
1. **Issues in the theoretical derivations.**
   I believe several of the demonstrations contain conceptual or mathematical inaccuracies:
   - **Appendix B.1:** Since $z$ is a continuous variable, $H(r)$ represents a *differential entropy*, which can be negative. This invalidates the step in the proof that relies on $H(r) \ge 0$.
   - **Appendix C.1:** The assumption that $g$ is invertible seems unjustified. As stated in the paper, the representations are *compact versions* of the inputs; thus, by definition, they are less informative than $x$. Consequently, there cannot exist an invertible mapping from $z$ to $x$.
   - **Equation (18):** The derivation implicitly assumes that the entropy of a continuous variable is always positive, which is incorrect. As a general rule, to establish lower or upper bounds in variational inference, it is preferable to work with quantities such as the KL divergence or mutual information, which are guaranteed to be non-negative, rather than entropy itself.

2. **Inadequate representation analysis (Section 4.1).**
   The evaluation relies mainly on t-SNE visualizations, which are qualitative and can be misleading. More appropriate metrics should be provided, such as the mean angle between embeddings of different classes or the intra-class concentration of embeddings. If the proposed regularizer is working as intended, the embedding space might actually appear *less clustered*, which is not necessarily undesirable.

3. **Redundant experimental sections (Sections 4.3 and 4.4).**
   These sections seem to evaluate closely related phenomena and could be merged to streamline the presentation.

4. **Overuse of appendices (Section 4.5).**
   Many essential results are relegated to the appendices. The main text should include the key experiments that substantiate the claims, while the appendices should only contain complementary or additional analyses.

5. **Missing calibration metrics (Section 4.5).**
   Standard metrics such as *Expected Calibration Error (ECE)* or *cross-entropy* would provide a more quantitative measure of how well the model’s predictive uncertainty is calibrated.

6. **Lack of comparison with standard uncertainty baselines.**
   The paper omits comparisons against established methods such as *ensembles*, *Monte Carlo dropout*, or *temperature scaling*. Including these baselines would contextualize the benefits of the proposed probabilistic regularization.

---

## Minor
1. In Equation (2), $z'_i$ and $z'_j$ are not defined.
2. In Equation (3), the subscript $n$ may cause confusion with “negative”; consider using $k$ instead.
3. The temperature parameter, standard in contrastive objectives, is omitted throughout the paper.
4. After Equation (10), it would be helpful to briefly explain what a “critic” refers to in this context.
5. In Proposition 3.2, clarify what it means for a critic to be *optimal*.
6. Equation (17) holds only if the covariance matrix is diagonal; this condition is not stated in the text and should be made explicit.
7. The derivation of Equation (18) seems to follow from combining (12) and (13), not (12) and (17), as currently indicated.

**Questions:**

1. Why is Equation (16) justified by the Data Processing Inequality (DPI)? The connection is not immediate and should be formally demonstrated.
2. Is the *projected normal* distribution essential for the approach, or could a simpler reparameterization (e.g., von Mises–Fisher) achieve comparable results?
3. Prior works such as [1,2] have used Gaussian encoders in contrastive learning, computing the KL divergence between *paired outputs* rather than against a prior. In practice, what do you see as the key differences between your approach and theirs?

**References**
[1] Federici, M., Dutta, A., Forré, P., Kushman, N., & Akata, Z. (2020). *Learning Robust Representations via Multi-View Information Bottleneck.* arXiv:2002.07017.
[2] Almudévar, A., Hernández-Lobato, J. M., Khurana, S., Marxer, R., & Ortega, A. (2025). *Aligning Multimodal Representations through an Information Bottleneck.* arXiv:2506.04870.

---

### Official Review · Reviewer_VR2y · 2025-10-30

**Soundness:** 2
**Presentation:** 2
**Contribution:** 2
**Rating:** 2
**Confidence:** 4

**Summary:**

The authors propose Variational Contrastive Learning (VCL), a decoder-free framework to equip contrastive learning methods like SimCLR and SupCon with a principled mechanism for uncertainty quantification, often matching or outperforming the accuracy of deterministic baselines. Two key components are introduced: the reinterpretation of the InfoNCE loss as a surrogate reconstruction term within the Evidence Lower Bound (or ELBO), and a KL divergence regularizer enforcing a uniform prior on the unit hypersphere. The framework maximizes this ELBO by modeling the approximate posterior distribution of the embedding given the input as a projected normal distribution, which enables the sampling of probabilistic embeddings. For implementation, the authors propose two instantiations, VSimCLR and VSupCon, which replace deterministic embeddings with samples from this posterior and add a normalized KL divergence term to the standard CL loss. The experimental results on multiple benchmarks show that VCL mitigates dimensional collapse, enhances mutual information with class labels, and matches or outperforms deterministic baselines in classification, with meaningful uncertainty estimates provided.

**Strengths:**

The paper provides a coherent theoretical reinterpretation of contrastive learning through a probabilistic lens. By formulating InfoNCE as a surrogate reconstruction term in an ELBO objective, it bridges a gap between variational inference and contrastive learning, offering a fresh theoretical grounding that could allow for further analytical work in this area.

The introduction of a uniform spherical prior and projected normal posterior shows careful consideration of embedding geometry. This design choice ensures compatibility with normalized contrastive embeddings and highlights the authors’ awareness of the geometric constraints underlying modern contrastive learning objectives.

Overall, the paper includes a nontrivial generalization bound for the KL regularizer, which complements the primarily empirical literature on contrastive learning with a theoretically grounded contribution that enhances understanding of probabilistic regularization effects.

**Weaknesses:**

1. The paper contains several flaws in mathematical derivation. While the experimental framework remains sound, these issues require correction to ensure the theoretical contributions align with the stated goals of providing a rigorous connection between InfoNCE and ELBO.

- **Incorrect sign propagation in the InfoNCE-ELBO connection**

  The paper's central theoretical contribution, minimizing InfoNCE asymptotically maximizes the ELBO reconstruction term, is incorrectly derived. In the proof of Proposition 3.2 (Eq. 30, Appendix B.2), the authors write $\lim_{N \to \infty}\{\mathrm{I_{NCE}}+\log N\}$ when it should be $\lim_{N \to \infty}\{-\mathrm{I_{NCE}}+\log N\}$, dropping a critical negative sign. This error cascades through Eq. 32, where the subtraction $\mathbb{E}[\log p(z'|z)] - \mathbb{E}[\log p(z')]$ is incorrectly written with a plus sign. As a result, the final expression in Eq. 13 shows the KL divergence and entropy terms being subtracted when they should be added.

  This fundamentally changes the theoretical interpretation: rather than InfoNCE maximizing the reconstruction term as claimed, the correct derivation shows InfoNCE provides a decomposition that adds regularization terms. To address this, the authors should consider: (1) restoring the negative sign in Eq. 30, (2) correcting the sign in Eq. 32, (3) revising Eq. 13 to show $\mathbb{E}[\log p(z'|z)] + D(q_\theta(z'|x)\Vert p(z')) + H(q_\theta(z'|x))$, and (4) updating claims made in the paper to reflect that InfoNCE provides a decomposition rather than directly maximizing reconstruction.

- **Inconsistent conditioning in independence assumptions**

  In Proof of Lemma 3.1 (Appendix B.1, Eq. 25), the Jensen inequality is applied using $p(z'|x)$ in the numerator, but the subsequent derivation assumes $p(z'|z)$. This contradicts the conditional independence assumption $x \perp z | z'$ that determines the entire theoretical framework. Since this lemma motivates the reconstruction bounds in Eqs. 10-12, the error propagates to the main theoretical results. The authors should replace $p(z'|x)$ with $p(z'|z)$ in Eq. 25 to maintain consistency with their independence assumptions and ensure the validity of the reconstruction lower bound.


2. Uncertainty evaluation is not aligned with the standard approaches.
The paper argues that the probabilistic embeddings provide useful uncertainty, but the evidence is largely correlational (e.g., log det K versus human label entropy on CIFAR-10H and corruption severity on CIFAR-10C, along with a small gain from log det K–weighted sampling). To support the claim that the method yields *useful* uncertainty, at minimum the authors should report negative log-likelihood (NLL), a strictly proper scoring rule for probabilistic predictions, together with Expected Calibration Error (ECE) and a reliability diagram to assess calibration. Stronger and increasingly standard practice would also include selective prediction (risk–coverage/AURC) and OOD detection (AUROC). These metrics are well-established in the literature: see [1] for why NLL is an appropriate scoring rule, [2] and [3] for calibration under modern deep networks, and [4] and [5] for uncertainty and distribution-shift evaluations.


3. The evaluation is confined to small/medium-scale datasets (CIFAR‑10/100, STL‑10, Tiny‑ImageNet, Caltech‑256) with a primarily ResNet‑18 backbone. These choices (as stated in Sec. 4 and Appendix D.1) make it hard to assess whether the method scales or competes under the standard large‑scale settings, typically ImageNet‑1000 with stronger backbones (ResNet‑50/ResNet-101/ViT), and beyond linear probing to retrieval and cross‑dataset transfer. To substantiate claims of broad applicability, I recommend adding at least one ImageNet‑1000 evaluation with stronger backbones, including momentum/queue‑based baselines under matched compute and comparable hyperparameter search, and reporting retrieval and transfer results alongside linear probe performance.

4. Table 1 indicates that the variational variants are not uniformly stronger than their deterministic counterparts: for example, VSupCon underperforms SupCon on Tiny-ImageNet (Top-1 48.30 vs 57.60) and Caltech-256 (83.06 vs 87.06), and VSimCLR underperforms SimCLR on Tiny-ImageNet (37.70 vs 38.95) and STL-10 (60.11 vs 60.44). The paper hypothesizes two causes, (i) objective mismatch (VSimCLR's objective aligns with Eq. (19) whereas VSupCon's does not), and (ii) task conflict (SupCon directly optimizes for classification, so adding a KL regularizer can hurt accuracy), and provides a small ablation suggesting that varying the KL (smaller $\beta$) weight brings about performance changes of VSupCon. I recommend reporting explicitly on the accuracy–uncertainty trade-off, especially for the large drops on Tiny-ImageNet and Caltech-256.

5. The SimCLR/SupCon figures in Table 1 appear to be in-house re-implementations under the paper's unified, small-scale protocol (e.g., ResNet-18, 32×32 inputs, batch = 512, $\tau = 0.07$, no momentum encoder/queue, all negatives in-batch, 500-epoch pretraining, AdamW, and linear probing), rather than the larger-scale configurations commonly used in the original works. This likely explains discrepancies with widely cited results, but it is not clearly flagged next to Table 1, which may confuse readers about cross-paper comparability. Please (i) state explicitly by the table that baselines are reproduced under your protocol (with a pointer to the training details), (ii) report mean ± std over multiple runs for Table 1, and (iii) where feasible, add a matched-protocol comparison to standard settings (e.g., ResNet-50/ViT, batch size of 1024 or even larger, and momentum/queue baselines) or cite the original numbers alongside your reproductions to uphold fair comparisons.



**References**
[1] Gneiting, Tilmann, and Adrian E. Raftery. *“Strictly Proper Scoring Rules, Prediction, and Estimation.”* Journal of the American Statistical Association, 2007.

[2] Guo, Chuan, et al. *“On Calibration of Modern Neural Networks.”* Proceedings of the 34th International Conference on Machine Learning (ICML). 2017.

[3] Minderer, Matthias, et al. *“Revisiting the Calibration of Modern Neural Networks.”* Advances in Neural Information Processing Systems (NeurIPS), 2021.

[4] Lakshminarayanan, Balaji, Alexander Pritzel, and Charles Blundell. *“Simple and Scalable Predictive Uncertainty Estimation Using Deep Ensembles.”* Advances in Neural Information Processing Systems (NeurIPS), 2017.

[5] Ovadia, Yaniv, et al. *“Can You Trust Your Model’s Uncertainty? Evaluating Predictive Uncertainty under Dataset Shift.”* Advances in Neural Information Processing Systems (NeurIPS), 2019.

**Questions:**

1. Could you provide a updated version of the proof for Proposition 3.2 with the proper signs throughout? Specifically, how does the corrected asymptotic limit (with the KL and entropy terms added rather than subtracted) affect your interpretation of the InfoNCE-ELBO connection?

2. What is the concrete necessity and added value of Variational Contrastive Learning (VCL) over deterministic CL? The paper motivates VCL conceptually, but it remains unclear why the variational treatment is needed in practice under standard CL pipelines. In rebuttal, please articulate: (i) the specific limitation(s) of SimCLR/SupCon that VCL uniquely addresses; (ii) the target metric(s) where VCL should be expected to outperform (e.g., NLL/ECE calibration, selective prediction, OOD detection, collapse mitigation) and under what conditions; and (iii) a minimal ablation that isolates the benefit, e.g., remove KL and posterior randomness while holding temperature, normalization, and augmentations fixed, and report the improved performance on the above metrics. A concise “accuracy–uncertainty” Pareto curve (deterministic vs VCL) on one dataset would make the case much clearer.

3. Have you evaluated whether the learned uncertainties transfer across datasets or tasks, or are they specific to the training distribution? Given the computational overhead of sampling during inference, what is the practical runtime comparison between your variational methods and their deterministic counterparts, especially for large-scale deployment?

---

### Official Review · Reviewer_o9fV · 2025-11-04

**Soundness:** 1
**Presentation:** 2
**Contribution:** 2
**Rating:** 2
**Confidence:** 4

**Summary:**

This paper introduces Variational Contrastive Learning (VCL), a framework that replaces the deterministic embeddings used in traditional contrastive learning with probabilistic representations. By leveraging variational inference, VCL extends contrastive learning into a probabilistic setting. In this framework, the conventional reconstruction term in the variational objective is replaced with the InfoNCE loss, enabling a decoder-free architecture. Two variants of VCL are proposed—VSimCLR and VSupCon—which are evaluated on several benchmark datasets, including CIFAR-10, CIFAR-10C, CIFAR-100, STL-10, Tiny-ImageNet, and Caltech-256, and compared against the baseline methods SimCLR and SupCon. Experimental results demonstrate that VCL effectively mitigates dimensional collapse, enhances mutual information between learned representations and class labels, and improves classification accuracy in the latent space.

**Strengths:**

The proposed method is leads to easy-to-implement objective. The empirical results shows some improvement compared to conventional contrastive learning methodology, especially in mitigating collapse phenomena.

**Weaknesses:**

1. The motivation of this paper is not compelling. Why is that the conventional contrastive learning employs a deterministic map to represent each sample a limitation? What is the meaning of the "uncertainty" is a representation when there is no "true" representative point in the latent space? How does the quantified uncertainty affect the downstream tasks?

2. The key approximation result (12) is not fully justified. It is based on the approximation (10), which in turn is based on Lemma 3.1. However, Lemma 3.1 only provides a lower bound of the target expectation $E_{q_{\theta}(z|x)}[\log p(x|z)]$, hence InfoNCE surrogate in (10).
Furthermore, in the proof of Lemma 3.1, it is unclear how equality (a) follows from the RHS of the above line. Isn't it $p(z'|x)$, not $p(z'|x)$ that should appear in the first term? Finally, the assumption of Lemma 3.1 that $x$ and $z$ are conditionally independent given $z'$ means that given $z'$, knowing $z$ does not provide additional information on $x$. I don't think this assumption makes sense.

3. The discussion on tightness in Appendix C requires that $z'$ is a deterministic function of $x$. With this there's no "uncertainty" in representation.

4. In addition to (12), the regularization term is also based on the lower bounds (16) of the desired term (15). Combined with the lower bound (12), this yields is the actual objective function (19), which is just the InfoNCE regularized by the Gaussian KL divergence with a diagonal covariance matrix (with symmetrization). That latter is exactly the regularization term in VAE. Then what is the point of the uniform prior on the hypersphere and projected normal variational posterior? These probabilistic models, that take the majority of page 5, can be totally disposed of.

5. That being said, the claimed better performance of VCL may be due to the implicit/explicit regularization of the InfoNCE term, rather than the "uncertainty of representation." The training scheme (20) can be understood as $z = f_{\theta}(x')$, where $x'$ is a randomly perturbed sample $x$. This corresponds to a typical data augmentation scheme. The VAE regularization term is explicit. These regularization schemes may help mitigating collapse phenomena, as claimed, but other points, namely understanding embeddings through distributions, a probabilistic ELBO viewpoint beyond mutual information, and controllable embedding distributions, are vague and pointless. For instance, how can one control the embedding distributions? The uniform prior on the hypersphere is *not* used at all, and it appears that the independent normal (as VAE) is the only prior employed.

6. The generalization analysis seems peripheral. How does explain the methodology and the outcomes of the experiments?

**Questions:**

See the Weaknesses section.

---

### Note · Authors · 2025-11-28

I have read and agree with the venue's withdrawal policy on behalf of myself and my co-authors.